# Regulation of endocrine cell alternative splicing revealed by single-cell RNA sequencing in type 2 diabetes pathogenesis
Jin Wang [1,6] ✉, Shiyi Wen [1,6], Minqi Chen [2,6], Jiayi Xie [2], Xinhua Lou[3], Haihan Zhao[3], Yanming Chen[1,4], Meng Zhao [2] ✉ & Guojun Shi [1,4,5] ✉

The prevalent RNA alternative splicing (AS) contributes to molecular diversity, which has been demonstrated in cellular function regulation and disease pathogenesis. However, the contribution of AS in pancreatic islets during diabetes progression remains unclear. Here, we reanalyze the full-length single-cell RNA sequencing data from the deposited database to investigate AS regulation across human pancreatic endocrine cell types in non-diabetic (ND) and type 2 diabetic (T2D) individuals. Our analysis demonstrates the significant association between transcriptomic AS profiles and cell-type-specificity, which could be applied to distinguish the clustering of major endocrine cell types. Moreover, AS profiles are enabled to clearly define the mature subset of β-cells in healthy controls, which is completely lost in T2D. Further analysis reveals that RNA-binding proteins (RBPs), heterogeneous nuclear ribonucleoproteins (hnRNPs) and FXR1 family proteins are predicted to induce the functional impairment of β-cells through regulating AS profiles. Finally, trajectory analysis of endocrine cells suggests the β-cell identity shift through dedifferentiation and transdifferentiation of β-cells during the progression of T2D. Together, our study provides a mechanism for regulating β-cell functions and suggests the significant contribution of AS program during diabetes pathogenesis.

Type 2 diabetes (T2D) is a chronic metabolic disease characterized by insulin resistance and defective insulin secretion resulting from a loss of β-cell function and cell mass[1,2]. Alternative splicing (AS) refers to a process of producing different mRNA isoforms from one gene, contributing to transcriptomic and proteomic diversity[3], while more than 95% of the genes undergo alternative splicing, and some 50% of disease-related mutations influence splicing[4–6]. Interestingly, previous reports have demonstrated that AS was linked to obesity, insulin resistance, and diabetes[7–10].

Aberrantly spliced genes were identified to associate with diabetes[11–13]. It has been reported that the shorter human insulin receptor isoform can prevent downstream insulin signaling, which impairs β-cell survival[14].

Exon-skipped Glucokinase (GCK), which is required for glucose-6-phosphate (G6P) formation to regulate insulin secretion[15], relates to diabetes[16,17]. The aberrant splicing of *PAX4* impairs PAX4 to repress insulin and glucagon by targeting their gene promoters, which increases apoptosis in β-cells upon high glucose exposure[18]. The increase of VEGF165b, which results from the aberrant splicing of VEGF, may impair angiogenesis, and further impede wound healing in diabetes[19].

Splicing factors are essential in regulating RNA splicing by recognizing specific cis-regulatory sequences[20]. The expression levels of some spliceosome and RNA binding proteins (RBPs), which were significantly altered in insulin-resistant or T2D donors, were associated with the progress of the

[1]Department of Endocrinology & Metabolism, Medical Center for Comprehensive Weight Control, The Third Affiliated Hospital of Sun Yat-sen University, Guangzhou, Guangdong, China. [2]Key Laboratory of Stem Cells and Tissue Engineering, Zhongshan School of Medicine, Sun Yat-sen University, Ministry of Education, Guangzhou, Guangdong, China. [3]Zhongshan School of Medicine, Sun Yat-sen University, Guangzhou, Guangdong, China. [4]Guangdong Provincial Key Laboratory of Diabetology & Guangzhou Municipal Key Laboratory of Mechanistic and Translational Obesity Research, The Third Affiliated Hospital of Sun Yat-sen University, Guangzhou, Guangdong, China. [5]State Key Laboratory of Oncology in Southern China, Sun Yat-sen University Cancer Center, Guangzhou, Guangdong, China. [6]These authors contributed equally: Jin Wang, Shiyi Wen, Minqi Chen. ✉e-mail: jin.wang@stjude.org; zhaom38@mail.sysu.edu.cn; shigj6@mail.sysu.edu.cn

disease[21,22]. Splicing factors, *NOVA1*, *NOVA2*, *ELAVL4*, *PRFP8/PRP8*, etc., which encode splicing factors or major spliceosomes, regulate insulin secretion or β-cell survivals[10,23]. Several RBPs, such as hnRNPs, HuR, and LIN28, have also been linked to diabetes and diabetic complications[21]. These observations add alternative splicing regulators as a novel layer of regulation in diabetes.

Transcriptomic analysis of islets from diabetes-resistant and diabetes-susceptible obese mice revealed that AS events were linked to insulin secretion, which might be modulated by the 54 identified histones and chromatin modifiers[24]. A recent study discovered that pro-inflammatory cytokine treatment in islet cells induced changes in alternative splicing[25], which was associated with ~900 stimulus-specific splicing events with the majority of them involving a skipped exon[9]. Moreover, the skipped exon in HLA-II, which might be regulated by SRSF2, was confirmed in β-cells[9]. However, the transcriptomic splicing profiles remain poorly characterized in endocrine cells of T2D at the single-cell level.

Hyperglycemia-induced β-cell dedifferentiation or transdifferentiation is a well-recognized mechanism of β-cell failure in diabetes[2,26–29]. An "α-cell-like" shift of β-cells, recognized by the reduction of β-cell identity genes, increased immature or dedifferentiation genes associated with a progenitor-like state[30,31], and increased α-cell identity genes[26,32,33], which can ultimately lead to the defective insulin secretion. Transcriptional factors, metabolic regulators, epigenetic mechanisms, and microRNAs emerge for the interpretation of compromised β-cell in diabetes[30]. However, cellular dynamics of β-cell dedifferentiation and transdifferentiation were not well described in diabetes.

Previous transcriptomic analysis exclusively uncovered cell-type specific splicing events in endocrine cells based on bulk RNA-seq data. The contribution of RNA splicing to the diversity of endocrine cell types and the underlying regulatory mechanisms have not been investigated. In our study, we used the full-length single-cell RNA sequencing data of human pancreatic cells to identify cell-type-specific RNA splicing exons of major endocrine cells, and differential splicing events associated with T2D by comparing ND and T2D splicing profiles. We demonstrated that cell-type-specific splicing events, independent of their gene expression regulation, drove the clustering of major endocrine cell types. Altered splicing profiles in endocrine cells enable the definition of the compromised mature β-cell subset in T2D, which could be potentially regulated by RBPs, such as hnRNPs and the FXR family. Finally, pseudotime analysis at the single-cell level revealed the loss of β-cell identity in T2D, which was attributed to β-cell developmental trajectory reversion and deflection, indicating their dedifferentiation and transdifferentiation.

## Results
### Pancreatic endocrine cell types revealed by single-cell splicing profiles
To investigate cell type-specific alternative splicing of pancreatic islets, we used the two sets of single-cell RNA sequencing data based on full-length Smart-seq2 technology with higher sequencing depths and cell numbers among the candidate datasets after evaluating seven independent datasets[34–40] (Table S1), derived from two independent cohorts with no-diabetic and type 2 diabetic individuals, produced by The Jackson Laboratory and Regeneron Pharmaceuticals (denoted as Lawlor and Xin hereafter)[38,39]. The Lawlor dataset includes 1050 cells, while the Xin dataset includes 1600 cells, with the median of 1.53 million and 1.07 million read counts, respectively (Fig. S1a). In our analysis, 972 cells from the Lawlor dataset and 1474 cells from the Xin dataset passed the quality control (minimum of 2500 genes and maximum of 10,000 genes per cell) (Fig. S1b). We identified the median of 4435 and 1528 AS events in these high-quality cells from the Lawlor and the Xin dataset, respectively (Fig. S1c), and then assigned to the transcriptional cell types, including endocrine cell types (α-, β-, *INS/GCG*-, δ-cell, and PP), and exocrine cell types (acinar, stellate, ductal and endothelial cell), based on specific marker genes as described in two original work[38,39] (Fig. 1a, b, Fig. S2 and S3a, b). A cluster in the Lawlor dataset simultaneously expressing *INS* and *GCG* mRNA as described in

previous studies[41–44] annotated as *INS/GCG*-cells in this study (Fig. 1a and Fig. S2a), which displayed similar cell quality with other endocrine cell types (Fig. S2b–d). We then quantified the exon inclusion levels (ψ/Psi, percent spliced in) of over 30,000 cassette exons with sufficient exon connection reads (≥20) in the Lawlor and Xin dataset, respectively, using the Quantas pipeline[45,46]. We focused on the Lawlor dataset on account of its larger number of exons and integrated the results with the Xin dataset.

We performed clustering analysis of the Lawlor dataset based on RNA splicing profiles with *t*-distributed neighbor embedding (t-SNE) (Fig. 1c), and the major endocrine cells were completely apart from non-endocrine cell types which were not clearly defined (Fig. 1d). To examine the regulation of endocrine cells in T2D, we identified 10 clusters of endocrine cells based on RNA splicing profiles in the Lawlor dataset (Fig. 2a). For the Xin dataset, we identified 12 clusters (Fig. S3c, d). Compared to the transcriptional cell types based on the gene expression profiles, splicing profiles enable the separation of β- (clusters 1 to 4), α- (clusters 6 to 9), and *INS/GCG*- (cluster 5) cells (Fig. 2a–c). δ-cell and PP cannot be well separated from major endocrine cells may be attributed to their low cell numbers leading to much lower resolution (Fig. 2c). We observed similar results in the Xin dataset, in which RNA splicing profiles revealed major endocrine cell types in pancreatic islets of ND and T2D individuals (Fig. S3e). The Lawlor dataset showed much unequivocal clustering in comparison to the Xin dataset, possibly associated with its 1.4-fold higher depth of sequencing and 2.9-fold median of AS events, and longer sequencing read length than the Xin dataset (Fig. S1a, S1c and Table S1).

Exon inclusion levels (defined by ψ) drive the separation of the subpopulations in β- and α-cells (Fig. 2a, b), showing different AS events that contribute to distinct isoform expression profiles (Fig. 2d). For example, exon 2 of *SEC13* and exon 2 of *C7orf44* gene-splicing made cluster 1, a subpopulation of β-cells, separate from other clusters, not driven by *SEC13* and *C7orf44* gene expression (Fig. 2e). Exon 2 of *KARS* and exon 2 of *SBDSP1*, not gene expression, enable the separation of cluster 6, a subpopulation of α-cells (Fig. 2f). In the Xin dataset, gene splicing profiles can also be used to distinguish subsets (Fig. S3f).

Together, splicing profiles at single-cell levels enable us to determine the major endocrine cell types exhibiting subpopulations.

### Cell-type-specific splicing events in endocrine cells
We next identified the specific splicing events which determined the endocrine cell-specificity. We detected 1164, 1175, 1418, 1447, and 1508 specific splicing events in β-, α-, *INS/GCG*-, δ-cell, and PP of the Lawlor dataset, respectively (|Δ ψ| > 0.1 and adjusted *p* value < 0.05) (Fig. 3a). To unravel the potential function associated with endocrine cell type, we performed gene ontology (GO) analysis to depict pathway enrichment of genes containing cell-type-specific splicing events (Fig. 3b). We found that pathways related to "RNA splicing", "tRNA metabolic process" and "Microtubule organizing center part" are shared by multiple cell types (Fig. 3b), indicating that a considerable RNA splicing-mediated diversity is required for these genes related fundamental islet property. On the other hand, we found that specific pathways were enriched in certain cell types, such as "ER to Golgi vesicle-mediated transport", "Golgi vesicle transport", "Regulation of protein stability" and "Endoplasmic reticulum-Golgi intermediate compartment" in β-cells, "Regulation of mRNA metabolic process", "tRNA modification" and "Protein K63-linked deubiquitination" in *INS/GCG*-cells, "mRNA export from nucleus", and "Ciliary transition zone" in α-cells (Fig. 3b).

However, signature genes expression enriched in different pathways, such as "Translation initiation", "Protein targeting to ER" in β-cells, "Extracellular structure organization", and "Cell-cell adhesion mediated by integrin" in α-cells, and no significant pathways in *INS/GCG*-cells (Fig. 3c). To further validate the relationship of splicing abundance and transcriptional levels in endocrine cells, we next compared cell-type-specific RNA splicing genes (|Δ ψ| > 0.1 and adjusted *p* value < 0.05) and signature genes ( | log₂FoldChange | > 0.25 and adjusted *p* value < 0.05). We found 823, 707, 985, 986, 999 specific splicing events and 105, 1054, 14, 210, 197 signature

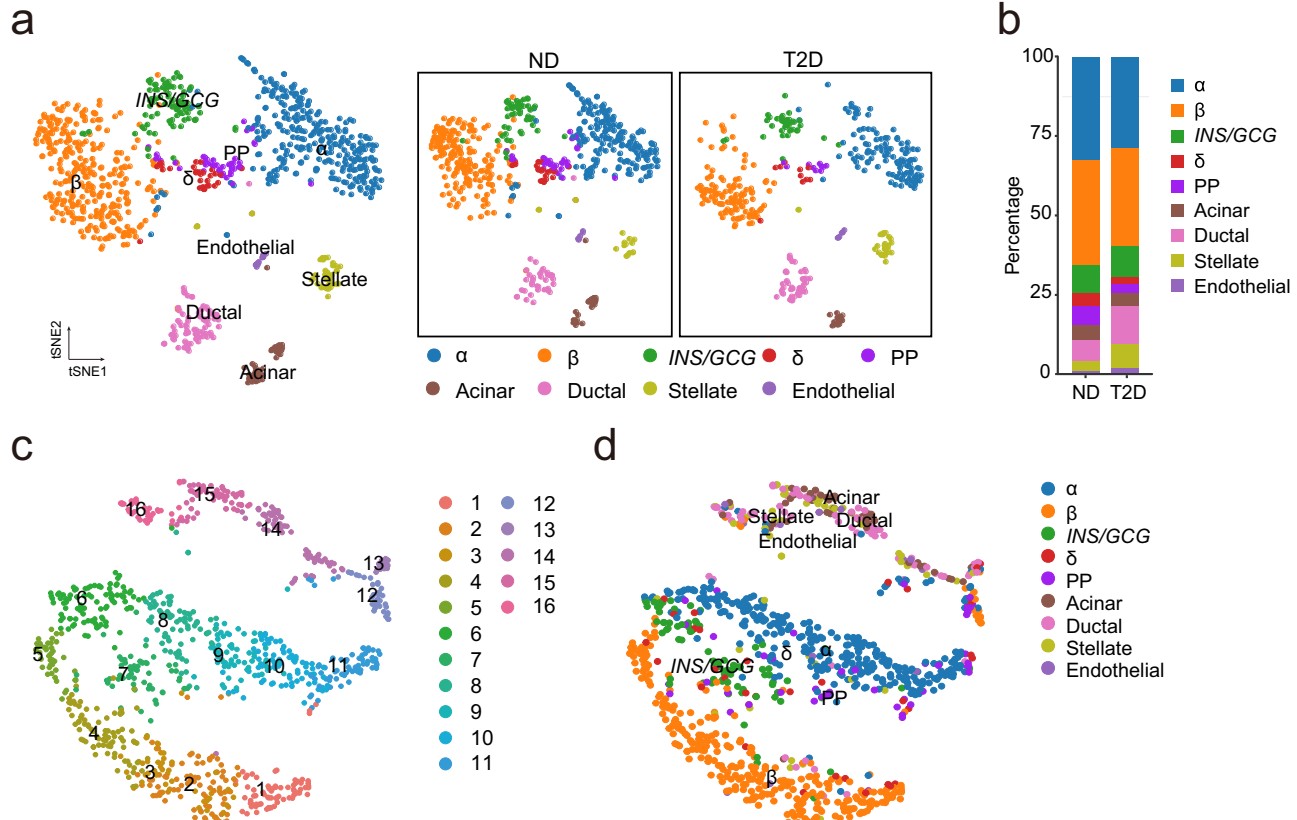

**Fig. 1 | The cell type annotation and splicing profiles of islet cells from ND and T2D individuals at the single-cell level. a** t-SNE plots of 972 ND (579 cell) and T2D (393 cells) islet cells based on gene expression from the Lawlor dataset. **b** Fraction of each cell type from ND and T2D islet cells. t-SNE plot clustered by splicing profiles from the Lawlor dataset (**c**), and cell types defined by gene expression profiles (**d**).

genes, sharing only 11, 117, 1, 41, 37 genes by specific splicing and signature genes in α-, β-, *INS/GCG-*, δ-cells and PP, respectively (Fig. 3d). As expected, the inclusion levels of these genes are not correlated with gene expression (Fig. 3e). For example, *SLC30A8* and *MRPS18C* enriched exon inclusion in α- and δ-cells but showed comparable gene expression levels in each cell type (Fig. 3f-i). We also observed the low correlation of cell-type specific splicing and gene expression in endocrine cells from the Xin dataset (Fig. S4). These results showed that cell-type-specific splicing events of each endocrine cell type are independent of gene expression, indicating an additional regulation layer of endocrine cell function.

### Differential splicing between ND and T2D endocrine cells

We have demonstrated that the splicing profile defined the cell-type specificity of endocrine cells in both ND and T2D (Fig. 2a–c and S5). In addition, ND and T2D β-cells shared only around 15% cell-type specific AS events through separating analysis (Fig. S6). AS acts as a prime source of transcriptional diversity which is thought to be associated with diabetes[1]. To evaluate the alternation of the transcriptional diversity in T2D, we next identified the differential splicing between ND and T2D endocrine cells. We detected 734 exons including events ($\Delta \psi > 0.1$ and adjusted $p$ value $< 0.05$) and 789 exon skipping events ($\Delta \psi < -0.1$ and adjusted $p$ value $< 0.05$) from 1030 genes in T2D β-cells compared to ND cells, as exemplified by the genes *SEC31A*, *GSTT1* and *ASNS* (Fig. 4a). Notably, we observed that genes that altered AS in T2D β-cells were significantly enriched in RNA splicing, centriole, regulation of mRNA metabolic process and mitochondrial protein processing (Fig. 4b), suggesting β-cell failure in T2D individuals[47].

To strengthen the credibility of the results, we integrated differential splicing between ND and T2D endocrine cells from two datasets, the Lawlor and Xin datasets. We discovered 178 AS events (74 as including events and 104 as skipping events in T2D compared to ND) from 153 genes showing

splicing differences in the same direction in two datasets (Fig. 4c). The genes of these common differential splicing events of these two datasets are enriched in "Clathrin vesicle coat" (3 genes), "Cilium assembly" (12 genes), "Regulation of RNA splicing" (13 genes) and "Regulation of vacuole organization" (7 genes) pathways (Fig. 4d), indicating the impaired insulin secretion of T2D β-cell.

Individual examples of differential splicing in β-cells in T2D compared to ND were discovered by differential analysis (Fig. 4e). We also identified seldom characterized examples, *APTX* and *EXOSC3* (Fig. 4e), which are important factors to maintain genome integrity[48] and play a role in RNA processing and degradation[49]. *APTX* exon 7 is highly included in T2D but skipped in ND β-cells, whereas *EXOSC3* exon 3 is skipped in T2D but included in ND β-cells (Fig. 4f, g).

Of note, only 8.99% of gene expression level changes in T2D were associated with AS (Fig. 4H). Moreover, the $\Delta \psi$ values of 178 differential splicing and their gene expression levels are not correlated (Spearman $\rho = 0.121$) (Fig. 4h). These suggest that splicing is an additional layer of regulation of endocrine cells by alternating transcriptional diversity during T2D, independent of gene expression levels.

To evaluate cell-type-specific differences, we also analyzed α- and *INS/GCG*-cells from ND and T2D islets in the Lawlor dataset. We identified 1688 differential splicing events (821 as including events and 867 as skipping events in T2D compared to ND) from 1129 genes and 999 differential splicing events (551 as including events and 685 as skipping events in T2D compared to ND) from 448 genes in α- and *INS/GCG*-cells, enriched for RNA splicing and centrosome organization pathways (Fig. S7), respectively.

Notably, nearly 50% (939 in α-cells, 789 in β-cells, 444 in *INS/GCG*-cells) of the differential splicing events are cell-type specific, while 794 (from 589 genes) and 150 (from 116 genes) differential splicing events are common in two or three cell types, respectively (Fig. S8a), indicating specifically

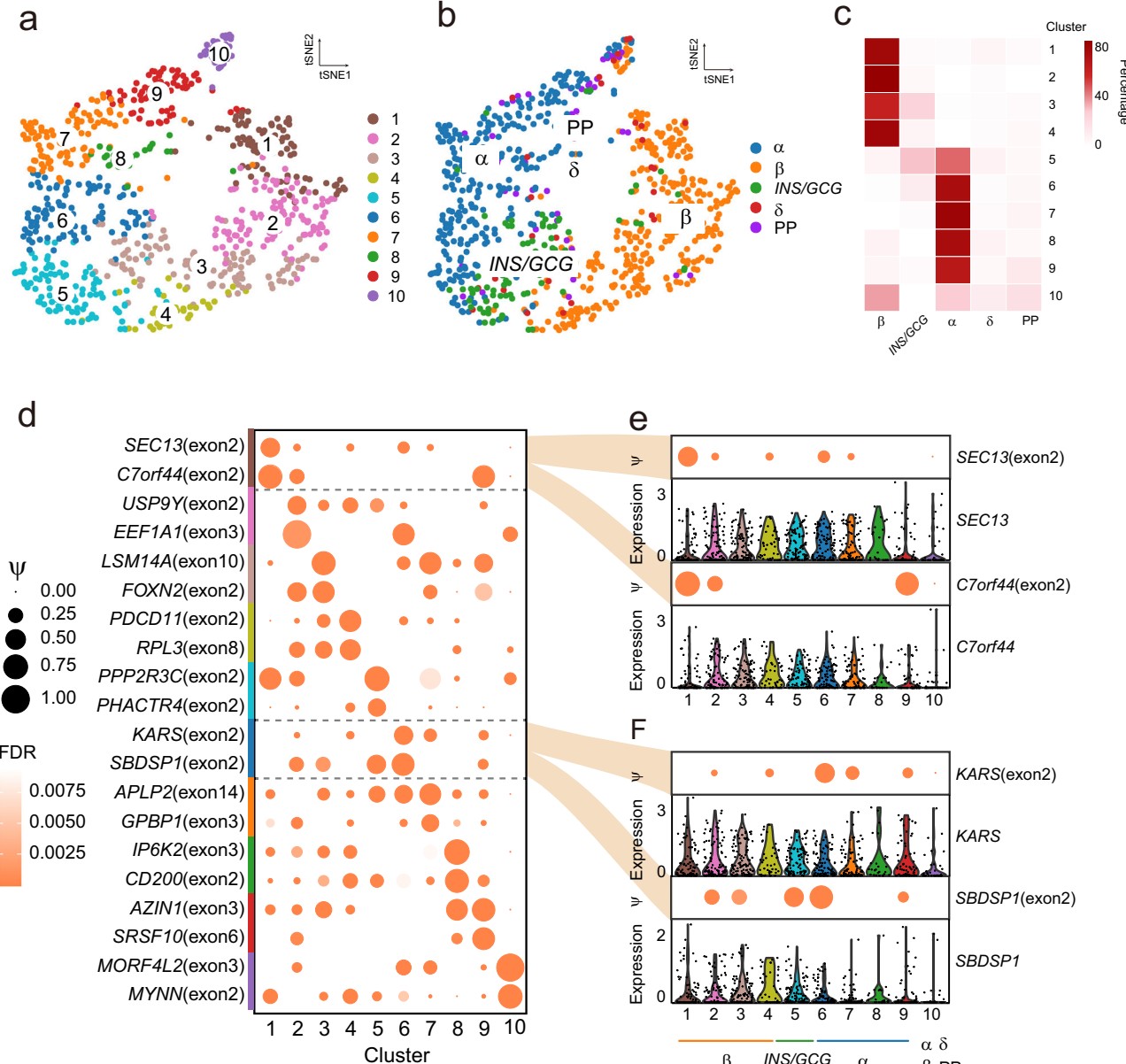

**Fig. 2 | Single-cell splicing profiles revealed α- and β-cells from ND and T2D individuals.** t-SNE plot showing 781 of ND and T2D endocrine cells from the Lawlor dataset. Cells are colored by cluster based on the splicing profiles (**a**), and endocrine cell types defined by gene expression profiles (**b**). **c** Heatmap showing the overlap between the clusters defined based on the splicing profiles and the cell types defined based on marker gene expression. The color key denotes the percentage of cells in each cluster defined based on the splicing profiles overlapped with the cell types. **d** Bubble plot showing inclusion levels (ψ) of specific splicing exons in each cluster based on splicing profiles. Dot size indicates the ψ value and color is the FDR value. Alternative splicing of gene *SEC13, C7orf44* (**e**) and *KARS, SBDSP1* (**f**) (bubble plot) and the gene expression level (violin plot) of each subpopulation within β- and α-cells.

splicing regulate different cell types of endocrine cells while sharing conserved splicing patterns of protein catabolic pathway genes in type 2 diabetes pathogenesis (Fig. S8b).

## RBPs regulate the skipping of splicing events in T2D β-cells
To examine how differential splicing between T2D and ND is regulated, we employed an unbiased de novo motif enrichment strategy to predict the potential splicing regulators. In our analysis, we employed an online web server rMAPS2 (RNA Map Analysis and Plotting Server 2)[50] to analyze differential alternative splicing to figure out potential cis-regulatory motifs in the cassette exons and their flanking introns (250 bp upstream or downstream of the target exon). This analysis identified 13 motifs (adjusted *p* value < 0.05) that are enriched in or around ND or T2D β-cell-specific exons in the Lawlor dataset. Seven motifs were identified for T2D β-cell-

specific included exons, corresponding to seven predicted RNA-binding proteins (RBPs) (*PCBP2, MBNL1, ESRP1, ANKHD1, FUS, SAMD4A, ZC3H10*) (Fig. S9A). MBNL1 and ESRP1 are inclined to bind to the cassette exon, whereas PCBP2, ANKHD1, and FUS bind to the upstream intron (Fig. S9a). These RBPs showed a slight increase (not significant) in T2D compared to ND β-cells (Fig. S9b), in line with the accumulated inclusion of their corresponding exons in T2D β-cells. Moreover, we found six identified motifs enriched in ND β-cell-specific included exons, corresponding to six predicted RBPs (*FXR1, HNRNPK, FMR1, SRSF10, HNRNPH1, LIN28A*) (Fig. 5a). hnRNPs accumulate at cassette exons (Fig. 5b), while FXR1 and FMR2 bind to flanking introns (Fig. 5c, d). We also observed the decreased expression of these RNPs in T2D compared to ND β-cells, especially *HNRNPH2* and *FXR1* which were significantly downregulated (Fig. 5e), consistent with the corresponding splicing enrichment pattern.

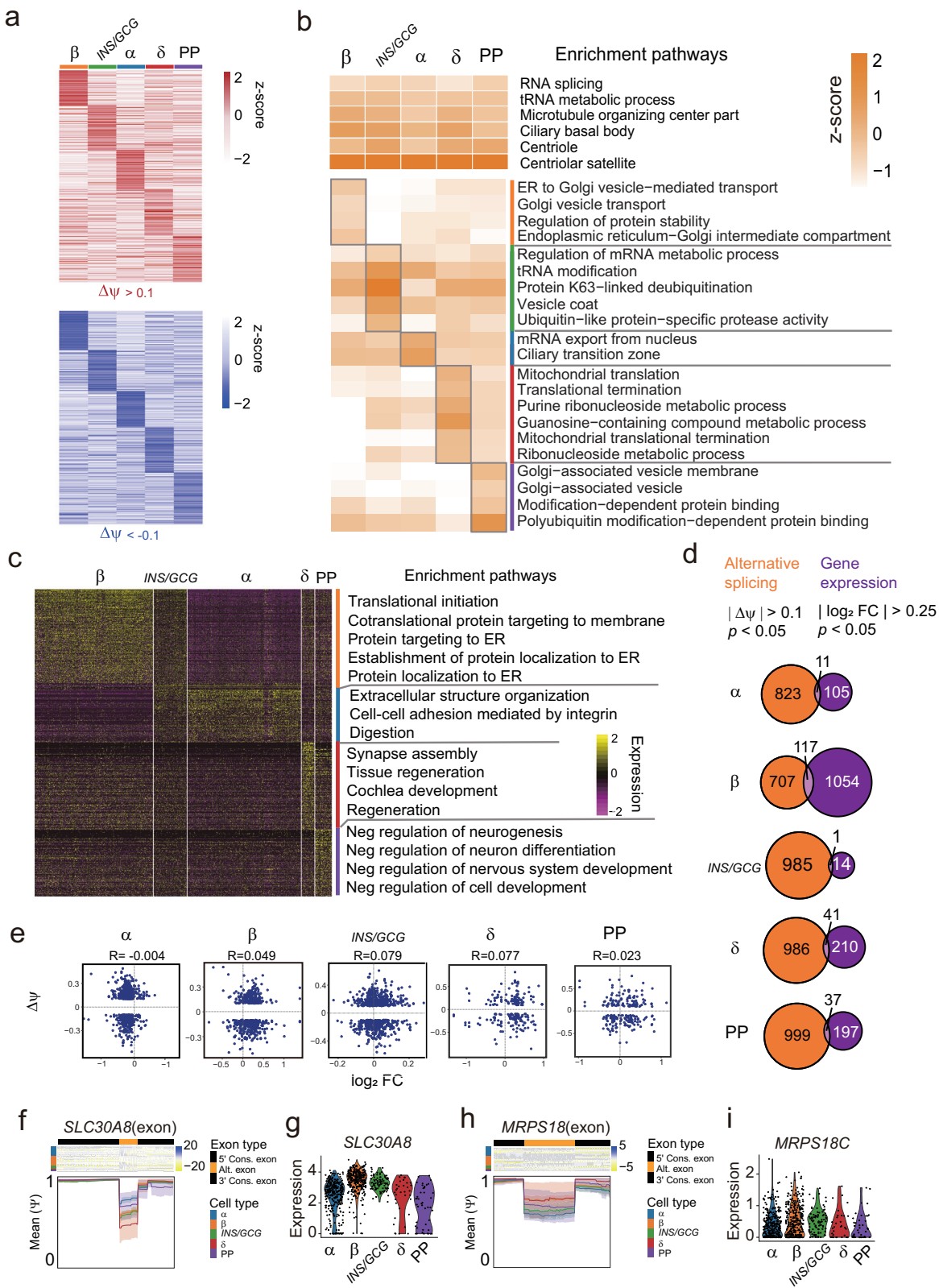

We found *HNRNPH2* was also significantly decreased in T2D β-cells of the Xin dataset (Fig. 5f). We then used predicted *hnRNPH2* and its regulated motif GGGAGGG in β-cells of the Lawlor dataset to blindly test whether *HNRNPH2* also exhibited evidence in regulation alternative splicing in β-cells of the Xin dataset. Further supporting the role of *HNRNPH2* in regulating splicing, exons containing motif GGGAGGG showed downregulated exon splicing in T2D compared to ND β-cells of the Xin dataset, as in the Lawlor dataset (Fig. 5g, h). hnRNPs exhibit similar overall expression profiles in both datasets (Fig. S10). Interestingly, we found other hnRNP family members, *HNRNPA1*, *HNRNPA2B1*, *HNRNPA1P10* were also downregulated in T2D β-cells of the Lawlor dataset (Fig. S10a), *HNRNPR* was downregulated in T2D β-cells of the Xin dataset (Fig. S10b).

**Fig. 3 | Cell-type-specific splicing events in endocrine cells. a** Heatmaps showing relative exon inclusion level (ψ) of specific exons in each endocrine cell type. Red and blue represent significant included (Δψ > 0.1 and adjusted *p* value < 0.05) and skipped (Δψ < -0.1 and adjusted *p* value < 0.05) in each cell type, respectively. Columns denote cells; rows denote splicing events. Z-score, row-scaled Δψ of the significant inclusion or skipping in each subpopulation. **b** GO analysis of cell-type-specific exons. Shared GO terms by multiple comparisons (significant in ≥ 3 comparisons) are shown at the top, while specific GO terms are shown at the bottom. **c** Heatmap of signature gene expression in endocrine cell subpopulations (fold-change > 0.25 and adjusted *p* value < 0.05) with GO analysis of signature genes listed on the right (color-coded by subpopulations). Columns denote cells; Z-score, rows denote genes. Row-scaled expression of the signature genes in each subpopulation scaled. **d** Overlapped gene numbers of cell-type-specific exons and cell-type signature genes in β-, α-, *INS/GCG*-, δ- and PP cells. **e** Scatter plots showing the association of cell-type-specific splicing genes and their gene expression in β-, α-, *INS/GCG*-, δ- and PP cells. Heatmap unraveling inclusion levels (ψ) (**f, h**) and violin plots representing gene expression levels (**g, i**) of *SRSF2* (**f, g**) and *RNH1* (**h, i**) gene. Mean ψ values across the genomic coordinates corresponding to the flanking constitutive exons and alternatively spliced exon. Shaded regions represent 95% confidence interval (CI) of the mean.

These results indicate a notion that hnRNPs might play pivotal roles in regulating splicing profiles in β-cells during T2D. Besides, exons containing motif ATGACA, which is regulated by *FXR1*, showed downregulated exon splicing in T2D compared to ND β-cells of the Lawlor dataset (Fig. 5i, j).

### RBP-mediated splicing alternation as a roadblock of β-cell maturation during T2D

β-cell functions and development are profoundly associated with diabetics. To describe the characteristics of β-cell during T2D at the single-cell level, we compared ND and T2D β-cells which were divided into major four subpopulations based on splicing profiles (Fig. 6a, b). By comparing the fractions of the four subpopulations, we found cluster 1 showed a threefold increase in T2D compared to ND β-cells, while cluster 4 profoundly decreased (Fig. 6b, c). The four clusters can also be defined by signature gene expression (Fig. 6d), albeit with fewer signature genes than cluster-specific AS events (Fig. S11, S12). We noticed that cluster 4 enriched genes in hormone secretion and insulin secretion pathways (Fig. 6e), indicating the functional impairment of β-cells might relate to exhausted cluster 4 in T2D. We found comparable *INS* expression in these four subpopulations, while *CHL1* expression, regulating insulin release[51,52], highly enriches in cluster 4 (Fig. 6f). Interestingly, *GLUT2* (*SLC2A2*), *MAFA*, *RFX6*, and *PDX1*, associated with cell maturation, are much more highly expressed in cluster 4 compared to clusters 1–3 (Fig. 6f). To quantify cell state in more detail, we started with a pseudotime analysis within β-cells using three methodologically distinct strategies in both the Lawlor dataset and the Xin dataset. We evaluated maturity scores based on β-cell developmental genes[53,54] (Fig. S13), while CytoTRACE estimated pseudotime based on gene diversity[55], and Monocle 2 revealed developmental trajectory based on analyzing changes in relative transcript counts[56]. Three assessments were highly correlated, showing higher maturity of cluster 4 compared to clusters 1 to 3 within β-cells in both datasets (Fig. 6g–i and Fig. S14a–f). In T2D, mature cells abruptly decreased in both datasets (Fig. 6j and Fig. S14g). Interestingly, we also validated immature and mature β-cells in another independent drop-seq dataset (denoted as the Fang dataset hereafter)[35] using these three methods described above and found the mature β-cells dramatically decreased in T2D (Fig. S14h–o). These findings indicate that the mature β-cell subset may play a key role in functional β-cells and blocking this mature subpopulation may result in insulin secretory failure leading to a progressive elevation in plasma glucose levels and diabetes.

To further explore alternative exons in regulating β-cell maturation, we next detected 936 and 807 exons as cluster 4 specific included and skipped exons (|Δ ψ| > 0.1 and adjusted *p* value < 0.05) from 688 and 620 genes within β-cells, respectively (Fig. 6k). As expected, we found *ADAL* exon was significantly included in cluster 4 (Fig. 6k) while skipped in T2D (Fig. 4a), and *GSTT1* exon was skipped in cluster 4 (Fig. 6k) while included in T2D (Fig. 4a). Moreover, we found cluster 4 included exons also enriched pathways, such as "RNA splicing", "Autophagy", "Chromatin modification", and "Cell cycle phase transition" (Fig. 6l and Fig. S15), which are identical to items enriched in T2D skipped exons (Fig. 4d). These indicated that the function of β-cells is attributed to the profile regulation of RNA splicing-associated cell maturation during T2D.

To discern regulators driving this process, we therefore, identified trans-regulators of splicing events enriched in cluster 4 using the online web server rMAPS2. This analysis predicted six RBPs that could bind to the enriched exons of cluster 4 or their flanking sequence (Table S2). As expected, we found FXR1 binds to the cassette exon and its downstream (Fig. 6m), and *FXR1* expression enriched in mature β-cells (Fig. 6n). We identified 79 exons or their flanking sequence from 70 genes containing potential FXR1 binding motifs across β-cells, with 29 of them described in previous report[57] (Fig. 6o and Fig. S16a, b). Gene expression levels of *FXR1* correlated with inclusion levels of these exons, suggesting that higher *FXR1* expression levels are linked to higher exon inclusion levels in mature β-cells, whereas lower *FXR1* expression levels with lower exon inclusion levels in immature cells (Fig. 6o and Fig. S16a). Interestingly, β-cell subsets exhibited cluster-specific targeted exon inclusion, indicating the diverse targets of FXR1 at different stages of β-cell maturation (Fig. 6o and Fig. S16a). Moreover, another FXR family member FMR1 and hnRNP members showed enriched gene expression in mature β-cells (Fig. 6n), which have been identified by comparing ND and T2D β-cells (Fig. 5). These results suggested that FXRs and hnRNPs link to β-cell maturation by regulating their RNA splicing profiles during T2D.

### Splicing impairment is associated with β-cell identity shift in T2D

The endocrine cell types were assigned based on the hormone expression levels (Fig. 1a and S2a)[39]. The previous results showed that insulin expression is comparable in mature and immature β-cells (Fig. 6f). It's impossible to use these alone to interrogate the identity shifts of endocrine cells. Therefore, we used published genesets[39,40,42,58] that reported to reflect β- and α-cell identities to score cell identity. By comparing ND β- and α-cells, we found scores of β-cells are much higher in clusters 1–4 (majority of β -cells) than clusters 6–9 (majority of α-cells), and the opposite is true for scores of α-cells (Fig. 7a, b), showing effective quantification of cell identities. In T2D, clusters 2 and 3 exhibited a significantly reduced β-cell score and a significantly increased α-cell score compared to ND clusters (Fig. 7a, b). We found β-cell enriched gene expression (*MAFA*, *CHL1*)[51,59] reduced and α-cell enriched gene *ARX* expression[60] increased in T2D compared to ND (Fig. 7c). These suggested that cluster 2 and 3, which were characterized as lower maturity of β-cell, were losing their β-cell identity, and shifting towards an "α-cell-like" phenotype.

Some evidence so far suggests that β-cell can dedifferentiate by reduced expression of β-cell identity genes[2,27,30], and transdifferentiate to α-cells by increased α-cell markers in diabetes[28,32,33]. To obtain cellular details in trajectory analysis, we employed RNA velocity, a kinetic model based on measuring transcriptional dynamics[61], to enable predictive cell fates. The directional flows of RNA velocity in ND β-cells exhibited consistency with maturity scores, CytoTACE analysis and Monocle 2 trajectory analysis (Figs. 7d and 6g–i). However, T2D β-cells showed β-cell dedifferentiation from cluster 2 to cluster 1, and transdifferentiation from cluster 3 to cluster 5 (*INS/GCG*-cells) or cluster 6 (α-cells) (Fig. 7d). Furthermore, developmental trajectory analysis by Monocle 2 confirmed the differentiation from cluster 1 to cluster 4 in ND β-cells, which exhibited different trajectory in T2D (Fig. 7e). The mature branch of β-cells was significantly decreased in T2D compared to those in ND, which we have observed in three independent datasets (the Lawlor dataset, the Xin dataset and the Fang dataset) (Fig. 7e and Fig. S14g, S14o). We also note that the terminal branch of the trajectory tree of T2D

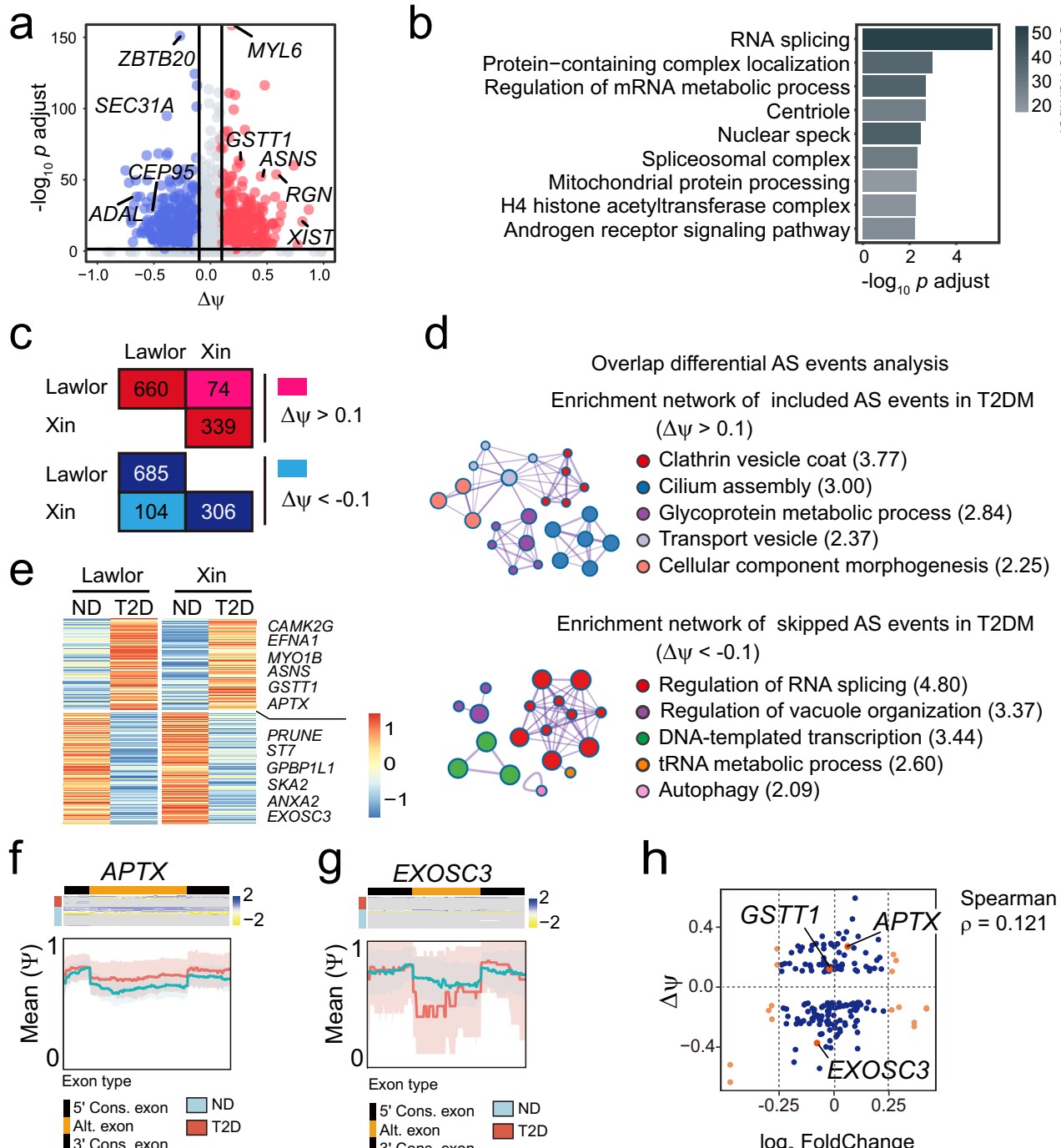

**Fig. 4 | Differential splicing events in β-cells between ND and T2D. a** Volcano plot showing differential splicing exons (|Δψ| > 0.1 and adjusted *p* value < 0.05) between ND and T2D β-cells. Red indicates exon included (Δψ > 0.1) and blue for exon skipped (Δψ < -0.1) in T2D β-cells. **b** GO analysis of differential splicing genes between ND and T2D β-cells. **c** Numbers of differential splicing exons with higher (red) or lower (blue) inclusion in T2D β-cells were detected in the Lawlor dataset and the Xin dataset. **d** Enrichment networks of common differential AS events in the Lawlor dataset and the Xin dataset showing in (**c**). Included (upper) or skipped (bottom) splicing genes in the T2D cells compared to ND. Number in bracket

indicates -log₁₀ *p* value. **e** Heatmap showing included or skipped splicing genes in T2D β-cells detected both in the Lawlor dataset and the Xin dataset. Examples of differential splicing genes *APTX* (**f**) and *EXOSC3* (**g**) showing included or skipped in T2D β-cells, respectively. Heatmap (upper) unraveling inclusion levels. Mean ψ values (down) across the genomic coordinates corresponding to the flanking constitutive exons and alternatively spliced exon. Shaded regions represent 95% confidence interval of the mean. **h** Scatter plot showing the association of signature genes fold change and Δψ in β-cells.

β-cells enriched more cells of cluster 1 than cluster 2 (Fig. 7e), suggesting the dedifferentiation from cluster 2 to cluster 1 as described in RNA velocity (Fig. 7d). As expected, we observed marker genes of dedifferentiated β-cells, *CD81* and *HES1*[31,62], were generally increased in β-cells (Fig. 7f), while maturation genes, *RFX6*, *CREB1* and *PFKFB2*[63], showed a

decrease (Fig. 7g). Moreover, clusters 5 and 6 in T2D showed increased β-cell scores, but not α-cell scores, may be attributed to their transdifferentiation from cluster (Fig. 7a, b). Cluster 5, which enriched the dual hormonal cells, showed bidirectional flows to both β- and α-cells in ND individuals, while a biased streamline towards α-cells in T2D (Fig. 7d,

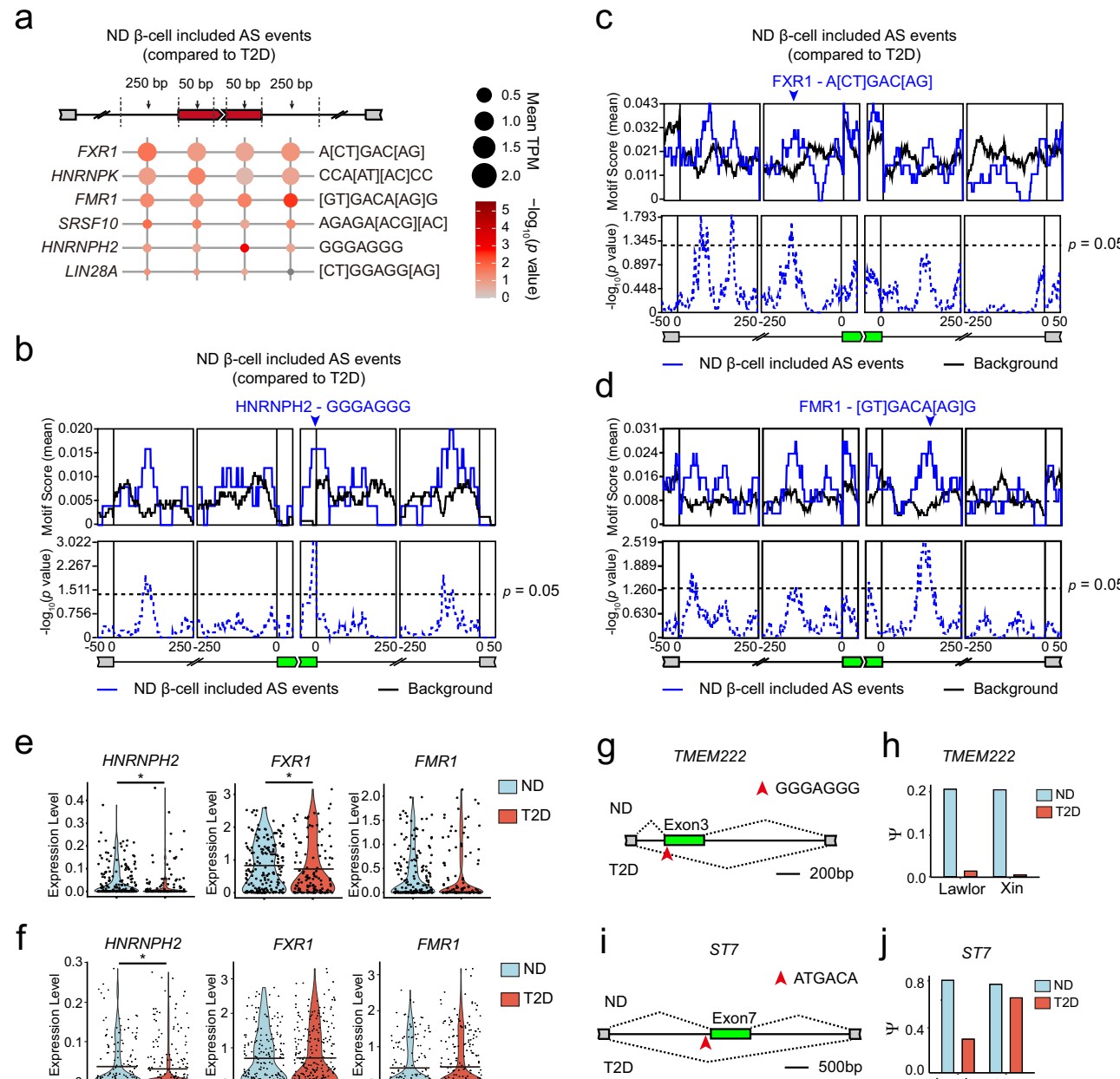

**Fig. 5 | Decreased RBPs regulate the skipping of splicing events in T2D β-cells.** **a** Bubble plot showing the predicted splicing factors from unbiased motif analysis of skipping AS events of T2D β-cells compared to ND β-cells in the Lawlor dataset. Motif sequences of corresponding splicing factors were labeled on the right. The dot color represents the smallest $p$ value in each enriched region, while the dot size indicates the median expression level of the splicing factors in ND β-cells. TPM, transcripts per million reads. Positional distribution of HNRNPH2- (**b**), FXR1- (**c**), and FMR1- (**d**) binding motifs of skipping AS events in T2D β-cells. Motif enrichment scores (top, solid line) and $p$ values (bottom, dashed line) were plotted according to AS event positions. Arrows indicate peaks of enrichment for exons. Violin plots representing the expression of indicated splicing factors in T2D compared to ND β-cells in the Lawlor dataset (**e**) and Xin dataset (**f**). **g** An example gene of *HNRNPH2* targeted exons with a binding motif upstream of the exon. **h** Inclusion levels of *TMEM222* (exon3) in ND compared to T2D β-cells. **i** An example gene of *FXR1* targeted exons with a binding motif upstream of the exon. **j** Inclusion levels of *ST7* (exon7) in ND compared to T2D β-cells. Lines in (**e**, **f**) indicate the mean gene expression. A two-sample KS test was performed to assess statistically significant (**e**, **f**), * $p < 0.05$.

down), indicating these dual hormonal cells may contribute to the compensation for the loss of β-cell mass, which impaired in T2D pathogenesis.

## Discussion

Previous studies using single-cell RNA-seq analysis of the pancreatic islets aimed to annotate new cell types or subtypes and to refine cell-type specific genes. Transcriptomics from endocrine cells profiled during development and disease benefits the community to extend our understanding of the roles

and functional status of endocrine cells in the pathogenesis of diabetes. However, the observations and predictions were mainly based on gene expression levels. Higher eukaryotes exhibit prevailing alternative splicing that plays a major role in expanding transcriptomic and proteomic complexity[64] to regulate the biological process, including the pathogenesis of obesity, insulin resistance, and diabetes[65–67]. Cell-specific splicing has been elucidated to define tissue compartments and cell types at the single-cell level[68,69]. In the current study, we characterized the genome-wide alternative splicing landscape of human pancreatic endocrine cells from ND and T2D

**Fig. 6 | A decreased subset of the mature β-cells in T2D links to RBP-mediated splicing profiles.**
**a**, **b** t-SNE plot of ND and T2D endocrine cells from the Lawlor dataset. Cells are colored by cluster based on the gene expression profiles (**a**) and splicing profiles annotated by diabetic conditions (**b**). **c** A neighborhood graph showing cluster enrichment and depletion in T2D compared to ND by Milo differential abundance testing. Nodes are neighborhoods, colored by their log fold change between T2D and ND. Nhood sizes correspond to the number of cells in each neighborhood. Graph edges depict the number of cells shared between neighborhoods. The layout of nodes is determined by the position of the neighborhood index cell in the t-SNE in (**b**). **d** Bubble plot showing signature genes of clusters 1 to 4. Dot size indicates the percentage expressed cells and color is the normalized average gene expression. **e** GO analysis of cluster 4 specific expressed genes compared to clusters 1–3. **f** Selected gene expression in clusters 1–4. **g** t-SNE plot showing maturity scores of clusters 1–4 overlaid with a contour map corresponding to maturity scores. The color indicates maturity scores. **h** CytoTRACE t-SNE plot of clusters 1–4. The color indicates the level of differentiation from low (blue) to high (red). **i** The pseudotime trajectory of clusters 1–4 (left) and their distribution over the tree structure (right) by Monocle 2 using DDRTree. Arrows indicate the trajectory of pseudotime pathway. **j** Area plots showing the estimated cell proportion of clusters 1–4 from ND and T2D, respectively. **k** Volcano plot showing cluster 4 specific splicing exons ($|\Delta\psi| > 0.1$ and adjusted $p$ value < 0.05) compared to clusters 1 to 3. Red indicates exon included ($\Delta\psi > 0.1$) and blue for exon skipped ($\Delta\psi < -0.1$). **l** GO analysis of cluster 4 specific splicing genes shown in (**k**). **m** Positional distribution of FXR1-binding motif of including AS events in cluster 4 compared to clusters 1–3 based on rMAPS results. Motif enrichment scores (top, solid line) and $p$ values (bottom, dashed line) were plotted according to AS event positions. Arrows indicate peaks of enrichment for exons. **n** Violin plots representing the expression of indicated splicing factors in clusters 1–4. **o** Heatmap showing the inclusion level of exons potentially targeted by *FXR1* in β-cell subsets. Dot size indicates the percentage of *FXR1* expression in each cluster, and color is the z-scores of *FXR1* average expression (upper) and z-scores of ψ (down).

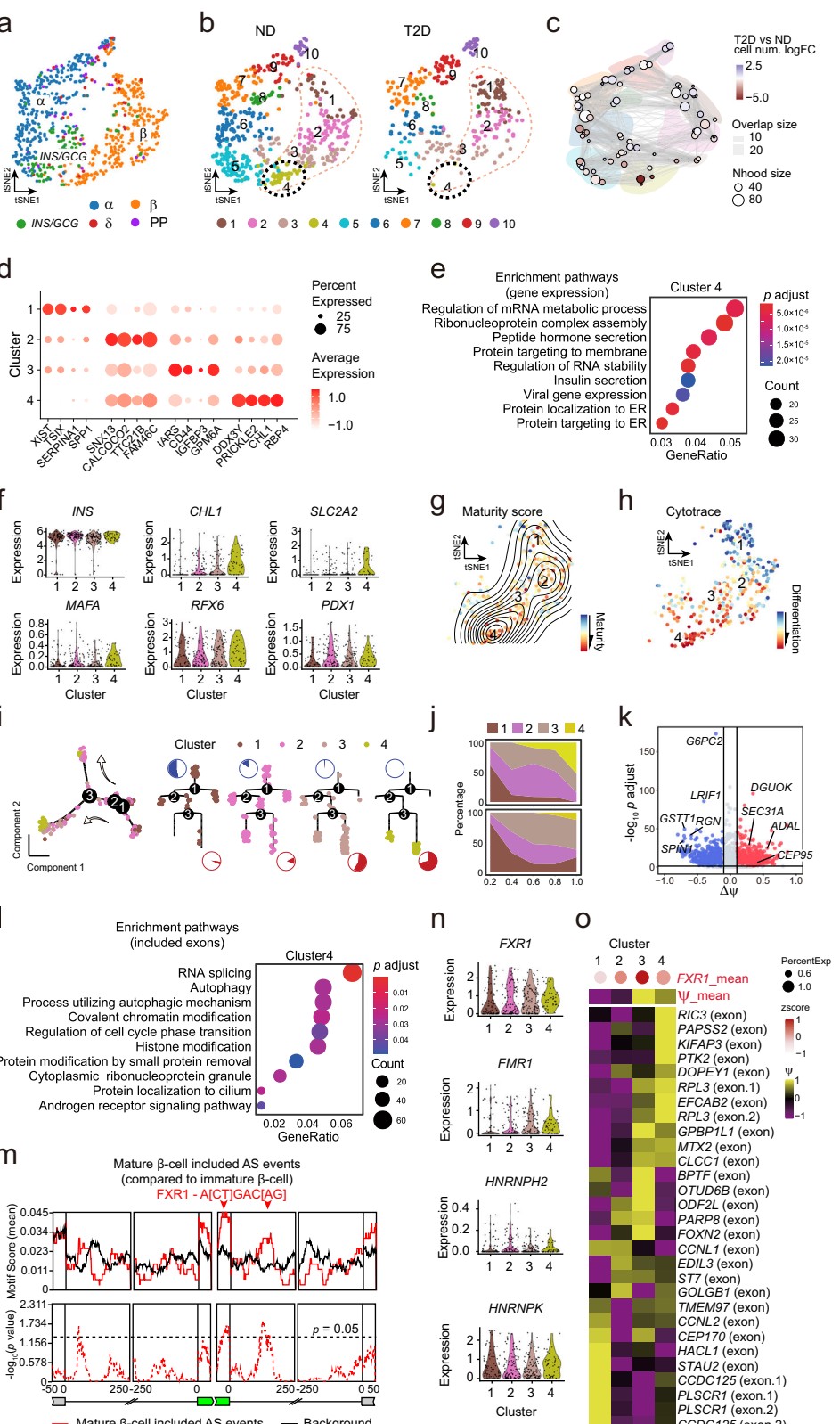

individuals at the single-cell level. We identified cell-type-specific and differential splicing events which were independent of gene expression regulations in ND and T2D endocrine cells. By comparing ND and T2D β-cells, we figured out that impaired RBP expression, such as hnRNPs and FXRs, may result in β-cell maturation arrests by altering their splicing profiles. Here, we investigated the gene splicing of human endocrine cells at the

single-cell level and potentially provided regulatory mechanisms and targets of β-cell fate determination during the pathogenesis of T2D.

In this study, we uncovered numerous specific splicing events in each cell type during T2D progression that occurred in the absence of appreciable changes at gene expression levels. For example, splicing events of "clathrin vesicle coat" pathway genes were illustrated to relate to T2D. These genes

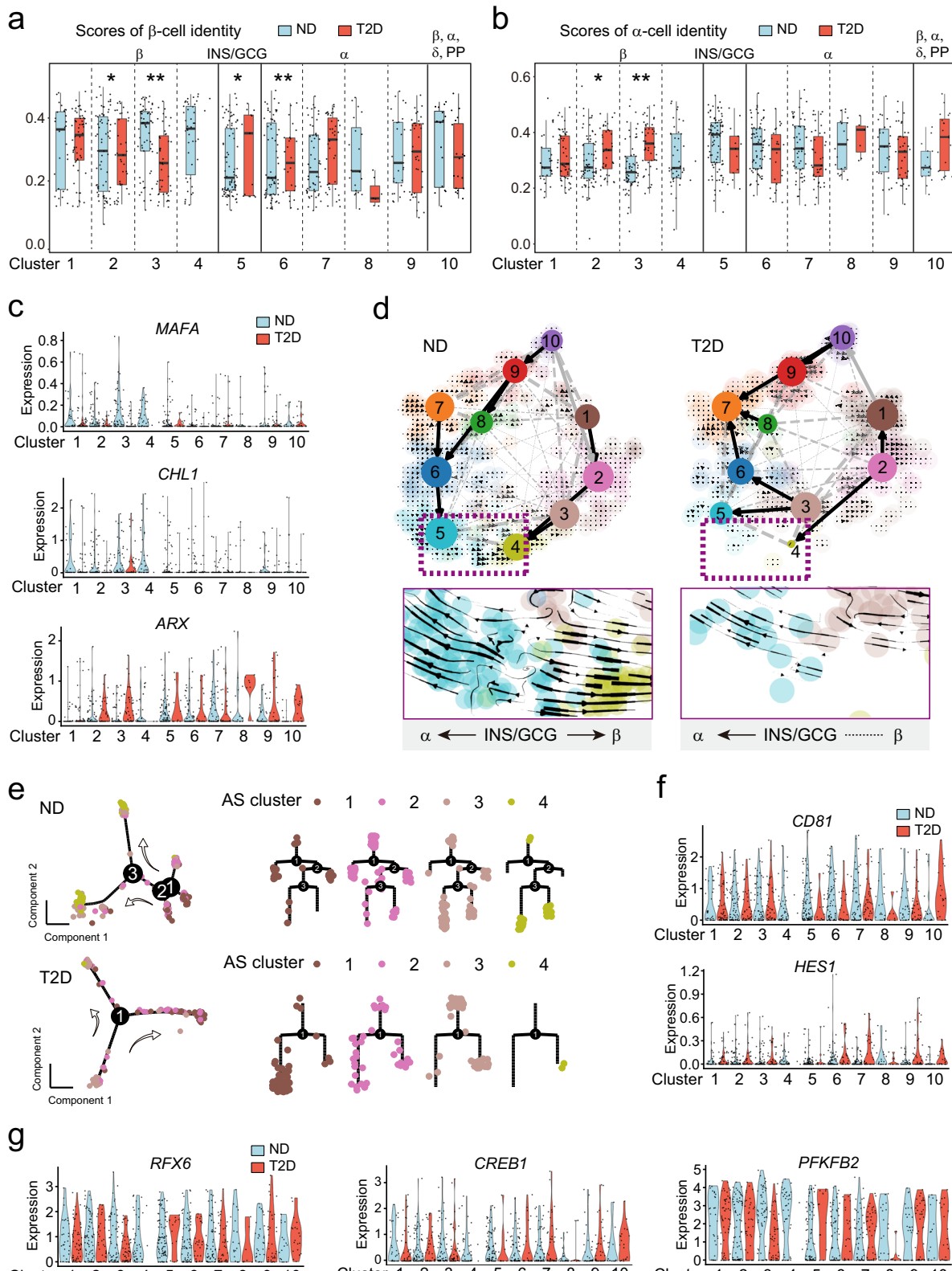

**Fig. 7 | Splicing impairment is associated with endocrine cell identity shift in T2D.** Scores of β-cell identity genes (**a**) and α-cell identity genes (**b**) of clusters 1–10 in ND and T2D endocrine cells. **c** Selected gene expression of β-cell identity and α-cell identity in clusters 1–10 from ND and T2D endocrine cells. **d** RNA-velocity analysis of ND and T2D endocrine cells with velocity arrows and corresponding principal arrows between subpopulations projected onto t-SNE plot of clusters 1–10. Purple boxes indicate the locations of regions magnified in the inserts with velocity

arrows. **e** The pseudotime trajectory for clusters 1–4 and their distribution over the tree structure in ND (upper) and T2D (down) by Monocle 2. Arrows indicate the trajectory of pseudotime pathway. Selected gene expression of β-cell dedifferentiation (**f**) and β-cell maturation (**g**) in clusters 1–10 from ND and T2D endocrine cells. A two-sample KS test was performed to assess statistical significance (**a**, **b**), * $p < 0.05$, ** $p < 0.01$.

play key roles in the regulation of (pro)insulin production and control insulin secretion in β-cells[70], and this process is precisely regulated by senescing glucose. Moreover, besides "RNA splicing" genes[25], skipped splicing events of "autophagy" pathway genes are associated with T2D, which impaired with the mature subpopulation depletion in T2D β-cells (Fig. 4d and Fig. S15b), confirming the findings from other reports showing the altered β-cell autophagic activity was implicated in T2D patients and rodents[71–73]. Interestingly, RNA alternative splicing was reported to regulate autophagy, especially in disease settings[74]. In addition, AS of "cell cycle phase transition" genes in the mature β-cells changed in T2D were consistent with the previous observation of cell cycle arrest in diabetes (Fig. S15b)[75]. Several RNA-binding proteins (RBPs), including hnRNPs, SRSFs, and HuD[76–78], especially hnRNPH2 significantly downregulated in both the Lawlor and the Xin dataset (Fig. 5e, f), have been identified as the regulators to guide the autophagic process in pancreatic hemostasis and tumors. hnRNPs were also reported to regulate insulin mRNA processing, translation, and insulin secretion in β-cells, and have been implicated with T2D[79]. Another group of RBP genes, such as *FXR1* and *FMR1*, were related to neurogenesis and cancers[80], while it remains unclear in regulating β-cell functions and diabetes despite the involvement of FMR1 in glucose homeostasis[81]. Accordingly, further functional assessment of their potential roles in regulating β-cells would benefit from their putative splicing program and targets.

Plastic β-cells undergo dedifferentiation in humans and mice with diabetes revealed by lineage tracing[27]. As apoptosis was thought to be over-evaluated during β-cell failure, dedifferentiation was supposed to be the major mechanism, thus, redifferentiation of the dedifferentiating or dedifferentiated β-cells to insulin-producing β-cells is expected to restore blood glucose levels[2,27]. By analyzing the dedifferentiating and transdifferentiating subsets of T2D β-cells, we identified AS alteration of "RNA splicing" and "mRNA process" genes may serve as common factors associated with T2D β-cell fate defect (Fig. S17). Interestingly, splicing of "nucleotide triphosphate metabolic process" and "mitochondrial inner membrane" would alter the β-cell determination through DNA synthesis and energy metabolism (Fig. S17)[78,82]. Combining the comparison of splicing in ND and T2D β-cell, our study through profiling RNA splicing at the single-cell level determined the dominant subset of c-cells for insulin secretion and predicted the developmental trajectory of dedifferentiation and transdifferentiation of β-cells in T2D associated with altered splicing diversity, energy status and cell cycle at AS regulation layers. This analysis provides sources and a paradigm to obtain novel therapeutic targets. Meanwhile, we revealed that RBPs, hnRNPs, and especially FXR family proteins as regulators of β-cell functions, along with various unidentified functions RBPs in different tissues. These findings would provoke more thoughts and investigations on the complex functions of RBPs and potential targeting strategies in diseases.

The activation of the nonsense mediated RNA decay (NMD) pathway, which could affect transcript levels, might be associated with proinflammatory levels in diabetes[83]. We investigated the NMD gene expression in the Lawlor dataset. We did not find significant differential expression between T2D and ND β-cells (Fig. S18). More experimental evidence remains needed to facilitate the understanding of NMD and T2D.

We incorporated diverse independent human datasets in our study; however, we refrained from directly combining these data because there were no appropriate normalization and batch effect correction methods for AS event profiles. Despite significant discrepancy among these datasets arising from the complex etiology of diabetes[84], we individually analyzed these datasets and integrated their results, identifying an observation and conclusion of commonality as the strategy described[46]. Specifically, AS events were found to reveal major types of endocrine cells, associated with hnRNPs and β-cell mature markers which exhibited similar gene expression profiles in both the Lawlor dataset and the Xin dataset (Fig. S10 and S13). Additionally, there was a common observation of β-cell fate determination arrest from three independent datasets in T2D by incorporating multiple trajectory analysis methods. Therefore, our analysis revealed consistently robust conclusions from datasets with great disparities. Nevertheless,

methodological development would largely improve the integration of diverse single-cell splicing data, thereby expanding sample sizes.

Besides cassette exons (or referred to as skipped-exons, SE) used in this study, we also included other six types of AS events, mutually exclusive exons (MXE), retained-introns (RI), alternative 5' and 3' splice sites (A5SS and A3SS), and alternative first and last exons (AFE and ALE) in our analysis using Quantas and MARVEL methods[85]. Both methods consistently identified SE as the predominant alternative splicing event, followed by MXE and A3SS as the second- and third-most prevalent types according to Quantas, and A3SS and AFE according to MARVEL (Fig. S19a, b). We also notice that identified A3SS are more than A5SS (Fig. S19a), possibly due to the 3' bias of smart-seq. Subsequently, we performed clustering based on these AS types (SE, MXE, A3SS. AFE), and observed that MXE, A3SS, AFE failed to clearly define the major endocrine cell types (Fig. S19c–h), We further included six AS event types with SE for clustering, but they did not significantly improve the clustering resolution compared to using SE only (Fig. S19i, j)[46], due to the lower AS event numbers of other six AS event types. So, it remains challenging to quantify adequate events of these AS types aiming for a higher resolution clustering at the single-cell level. In addition. SE contributes to the most influential factors in β-cells when identifying cell-specific AS events with the inclusion of seven AS types (Fig. S20). Therefore, we focused on the cassette exons/SE which represent the primary type of AS events[46].

In endocrine cells, cell-type-specific and differential splicing between ND and T2D were independent of their gene expression. This feature of transcriptomics was also observed in hematopoietic cells[86]. To separate major endocrine cell types, the gene expression and splicing profiles lead to the equifinality, which was emphasized in different tissues by independent studies using scRNA-seq[46,68,86]. Furthermore, an atlas across human tissues at the single-cell level revealed the fundamental rules for cell-type- or compartment-specificity of the alternative splicing programs[69]. Technically, bulks of tools have been developed for deep scRNA-seq data and 10x data[46,87–89]. Two sets of full-length Smart-seq2 data denoted as the Lawlor dataset and the Xin dataset have been used in our study and we noticed that the Lawlor dataset with 1.53 million reads and 128 thousand detected junctions per cell show more unequivocal clustering in comparison to the Xin dataset, with only 1.07 million reads and 73 thousand detected junctions per cell (Figure S1a-c). This observation indicates that deeper sequencing allows more detected junctions to reflect the more complete profiles of the splicing program at the single-cell level. We also note that the Xin dataset was captured with lower sequencing read length than the Lawlor dataset (Table S1). Thus, sequencing depth and read length, as well as sequencing mode, is worthy to take elaborate consideration and evaluation during splicing analysis. A recent study implemented long-read RNA-seq to directly analyze isoforms of transcripts and depicted the mRNA architecture as a dominant mechanism in regulating hematopoiesis[86]. Single-molecule sequencing will make appreciable advances in this field. Collectively, our study provides additional insights into the understanding of diabetes pathogenesis and transcriptomic alternative splicing. Our findings not only suggest that RNA architecture is an important step of gene regulation as well as transcriptional levels, but also provides alternative strategies for alleviating β-cell dysfunction and diabetes.

## Methods
### scRNA-seq data preprocessing
Islet cell RNA-seq data were mapped by OLego (v1.1.2)[90] to the reference genome (hg19). Only reads unambiguously mapped to the genome or exon junctions were used for downstream analysis.

### AS and gene expression quantification
We used the Quantas pipeline (http://zhanglab.c2b2.columbia.edu/index.php/Quantas) to quantify AS based on the number of exon junction reads, and only exon skipping events were analyzed in this study. The level of inclusion of alternative exons was represented by percent spliced in (PSI) or

ψ. We required exons to have junction read coverage ≥20 when estimated ψ. Gene expression quantification was quantified using the same pipeline.

For MARVEL pipeline[85], splice junction count matrix, intron count matrix, alternative splicing events, gene and sample metadata, normalized gene expression matrix and gene transfer file (GTF) were required as inputs. Then the gene metadata information was parsed and retrieved from gencode.v38lift37.annotation.gtf (https://www.gencodegenes.org). Trimmed reads were mapped to the hg19 reference genome using STAR v2.7.10b[91] in 2-pass mode. rMATS[92] was used to identify SE, MXE, RI, A5SS and A3SS splicing events using gencode.v38lift37.annotation.gtf. The intron coverage was computed using Bedtools[93]. MARVEL R objects were created by CreateMarvelObject function. AFE and ALE events were detected by DetectEvents function. We required splice junction reads ≥10 when estimated ψ.

### Unsupervised dimensionality reduction and hierarchical clustering of single cells using splicing profiles

Seurat v3[94] was used to analyze splicing and gene expression profiles. Cells expressing < 2500 or >10,000 genes were removed ($n = 74$ in Lawlor data and $n = 126$ in Xin data).

We next performed the t-SNE analysis of single cells based on the splicing profile of cassette exons in Lawlor's data. To filter unquantifiable exons, we maintained exons (4616 exons in Lawlor data and 2974 in Xin data passed this filtering) with junction read coverage ≥20 in ≥10% of 972 cells. The missing values in y matrix were replaced by an extract value far from all y. Then using the Bayesian principal component analysis (PCA) method to impute the remaining exons and the top PCs of the variance were further clustered using FindClusters, a shared nearest neighbor (SNN) based clustering algorithm, within Seurat to identify clusters (using top 10 PCs with a resolution of 1.2 for the Lawlor dataset, and top 10 PCs with a resolution of 1.2 for the Xin dataset) and using t-SNE for data visualization (Fig. 2). Similar analyses were repeated using only the endocrine cell.

Nine islet cell types have been assigned based on the expression of specific signature genes *INS* (β), *GCG* (α), *SST* (δ), *PPY* (PP/gamma), *GHRL* (epsilon), *PRSS1* (acinar), *COL1A1* (stellate) and *KRT19* (ductal)[39]. We used these marker genes to determine the representation of each islet cell type among our 972 single cells (Fig. 1). Then we counted the proportion of each cell type in ND and T2D, respectively (Fig. 1b) and calculated the fraction of each splicing cluster during cell types and patient status (Fig. 2c).

### Detection of differentially spliced exons

Differential splicing of cassette exons (DEs) was performed using the Quantas pipeline. For the Lawlor dataset, we performed one vs. other comparisons between each splicing cluster and defined the exons with the following criteria as significant: junction read coverage ≥ 20, False discovery rate (FDR, BH correction) ≤ 0.05, exon quantifiable in ≥ 10% of cells in different groups and $|\Delta\psi| \geq 0.1$. Alternative splicing marker events were defined by the top DEs after ranking by $|\Delta\psi|$. Differentially expressed genes were identified by Seurat. Gene with log2 FoldChange > 0.25 and adjusted $p$ value < 0.05 was reported as significant. For MARVEL, Anderson-Darling test was used for comparing the overall ψ distribution between two cell populations. Exons with adjusted $p$ value < 0.1 and outlier = FALSE were defined as differential splicing exons.

We also identified exons differentially spliced between T2D and ND, and between five endocrine islet cell types. IGV[95] and VALERIE[96] were used to visualize DEs specific splicing sites.

### Enrichment analysis

GO term enrichment analysis was performed on genes containing exons with differential splicing and differential expression genes (DEGs, calculated using a non-parametric Wilcoxon rank sum test with $p$ values adjusted using Bonferroni correction) between endocrine cell types, endocrine alternative splicing clusters, as well as T2D vs. ND in β-cells by clusterProfiler[97]. Heat maps were performed using the pheatmap package. The intersections between DEs and DEGs were identified using the Venndiagram package. Enrichment network was performed using the Metascape online web server[98].

with the "Express analysis" option. Differential abundancy testing between ND and T2D in the endocrine cells from the Lawlor dataset was performed with the R package MiloR[99]. To test for differential abundance, we used the QL method in edgeR to analyze neighborhood counts, and use the QL F-test with a specified contrast to compute a $p$ value for each neighborhood.

### De novo motif enrichment analysis and RBP prediction

De novo motif enrichment analysis and inferring RBP activity in regulating β-cells specific splicing exons was performed using rMAPS2[50] with default parameters. Sequences of significant differential or cell-type-specific splicing exons (junction read coverage ≥ 20, FDR ≤ 0.05, exon quantifiable in ≥10% of cells in different groups and $|\Delta\psi| \geq 0.1$) with unregulated exons (FDR > 0.05 or $|\Delta\psi| < 0.1$) set out as background events were analyzed. Enrichment motif and related RBP were defined as the minimum $p < 0.05$ in at least one of four regions (upstream intron, target exon 5', target exon 3', downstream intron), adding the maximum mean motif enrichment score > 0.01 in at least one of the four regions. The smallest $p$ value in each enriched region was adopted to generate the bubble plot.

### Pseudotime trajectory analysis

We implemented CytoTRACE analysis by inputting the normalized expression matrix to the CytoTRACE webtool (https://cytotrace.stanford.edu/).The output CytoTRACE score of each cell was then integrated with scRNA-seq data to be projected onto the UMAP.

We used RNA velocity to infer directionality of endocrine cell differentiation implementing the velocyto pipeline (version 0.17.17)[61]. In brief, spliced and unspliced reads were quantified and integrated with annotated scRNA-seq data using the scVelo python package (version 0.2.4)[100]. velocyto-drived counts were preprocessed, filtered (counts ≥20) and normalized based on 3000 highly variable genes. First-order and second-order moments for each cell were computed across its nearest neighbors. Velocities were estimated using a stochastic model of transcriptional dynamics with scVelo and projected onto UMAP embedding. Partition-based graph abstraction (PAGA) analysis was performed with Scanpy using default settings.

For the pseudotime trajectory analysis by Monocle 2[101], CellDataSet objects were created and the 'differentialGeneTest' function was used to derive DEGs of each cluster and pseudo-time analysis by genes with $q$ value < 0.01. Then the differentiation trajectory was inferred with the default parameters after dimension reduction and cell ordering.

Cell maturity and identity scores were evaluated with the AddModuleScore function of Seurat using published gene lists: *INS*, *SLC2A2*, *MAFA*, *RFX6*, *PDX1*, *CHL1*, *GCK*, *PPARGC1A*, *MDH1*, *NEUROD1*, *CREB1*, *G6PC2*, *PFKFB2*, *PFKM*, *SIX2*, *SIX3*, *ENTPD3*, *GPD2*, *DNMT3A*, *MTOR*[51,54,63,102–104].

### Statistics and reproducibility

For scRNA-seq data, the Lawlor dataset contains 5 ND and 3 T2D patients, the Xin dataset contains 12 ND and 6 T2D patients. A two-sample KS test was performed to assess statistical significance to compare means where appropriate. The threshold for statistical significance was p < 0.05. All correlations were calculated based on normalized expression values with the Spearman's Correlation.

### Reporting summary

Further information on research design is available in the Nature Portfolio Reporting Summary linked to this article.

### Data availability

Two single-cell sequencing FASTQ data produced by The Jackson Laboratory and Regeneron Pharmaceuticals were downloaded from NCBI Sequence Read Archive (SRA) under accession numbers SRP075970 and SRP075377, respectively. Another scRNA-seq FASTQ data by Department of Genetics and Genome Sciences was downloaded from NCBI SRA under accession number GSE101207. The source data behind the graphs in the paper can be found in the Supplementary Data 1[105].

## Code availability

The code used in the study is available from the Zenodo repository[106].

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

## Acknowledgements
We thank the National Natural Science Foundation of China (82070811, 82270886), Sci-Tech Research Development Program of Guangzhou City (202201020497, 202201020589, 2024A04J6567), China Postdoctoral Science Foundation (2021M693614).

## Author contributions
J.W., S.W. designed and performed computational analysis, and generated figures. M.C. contributed to computational analysis and generated figures. J.X., X.L and H.Z. provide discussion or editing of the manuscript. J.W., S.W., and G.S. wrote the manuscript. J.W., Y.C. and G.S. provided funding support. G.S., J.W., and M.Z. supervised the project.

## Competing interests
The authors declare no competing interests.
