## [Peer Review File · Communications Biology]

Reviewers' comments:

Reviewer #1 (Remarks to the Author):

Single-cell alternative splicing is an underappreciated area of investigation that has the potential to reveal disease and cell type-specific mechanism and biomarkers. The authors characterised the aberrant alternative splicing landscape of endocrine cells derived from type-2 diabetic (T2D) relative to non-T2D controls at single-cell resolution, with particular focus on the insulin-secreting β -cells. I particularly appreciated the demonstration of differentiation block in β -cells revealed through alternative splicing analysis but invisible at the gene expression level. I also appreciated the motif analysis to identify RNA-binding proteins whose motif were enriched among differentially spliced exon, which represents one step further post-differential splicing analysis. Overall, the narrative is concise and logical. While the study lacked experimental or functional validation (such as quantitative polymerase chain reaction, knock-in or knock-out/-down assays) of candidate alternative splicing events, this should not preclude the study from publication because the authors generated probable hypothesis on the disease mechanism and biomarkers from their analysis. However, it is exactly this same point, combined with the fact that majority of conclusions hinged on a single dataset (Lawlor et al.), that the computational analysis and validation should be ensured to be as robust as possible. To this end, I offer the following recommendations. I look forward to the author's response and the imminent publication of this study, provided satisfactory response.

Major:

1. Only skipped-exon (SE) alternative splicing (AS) events were analysed by the authors. Nevertheless, other AS event types have been shown to play important roles in health and disease states, namely mutually exclusive exons (MXE), retained intron (RI), alternative 5' splice site (A5SS), and alternative 3' splice site (A3SS). For example, PAK4 A3SS has been shown to dysregulate insulin and glucagon production (as cited by the authors in reference number 15). Moreover, two additional AS event types, namely alternative first and last exons (AFE, ALE), have been reviewed to be biologically relevant as well (Alvelos et al., Diabetes Obes Metab, 2018). In the interest of a more comprehensive characterisation of the AS landscape in T2D and also in identifying additional and novel biomarkers, the authors should include all seven AS event types in the analysis.
2. Quantas pipeline was used to detect, quantify, and perform differential splicing analysis. Quantas was first developed in 2013 originally for AS analysis of bulk RNAseq. Since then, several specialised single-cell AS analysis softwares were developed, benchmarked, and peer-reviewed. These published single-cell AS analysis softwares demonstrated their ability to

comprehensively characterise the single-cell AS landscape and to address challenges specific to AS analysis in single cells. One such published software is MARVEL

(<https://github.com/wenweixiong/MARVEL>; Wen et al., *Nucleic Acid Res*, 2023). It would be of particular importance to orthogonally validate the differentially spliced AS events detected by Quantas by identify AS events that are detected by both approaches and missed AS events uniquely detected by either approach.

3. Given the lack of experimental validation of candidate splicing AS events, it would be of particular importance to perform in silico visual-based validation of these AS events. The authors performed in silico validation in a pseudo-bulk manner, but this undermines the advantage of single-cell analysis (Fig4F and G). One such software that enables single-cell in silico validation of AS events is VALERIE (<https://github.com/wenweixiong/VALERIE>; Wen et al., *PLoS Comput Biol*, 2020). in silico validation will lend confidence for selection of true positive AS events for the postulation of disease mechanism and downstream experimental validation.

4. The β -cells differentiation block demonstrated by the authors were particularly intriguing and indicated a mechanism for T2D with therapeutic implications. The RNA velocity analysis was convincing, but cell numbers were nevertheless small, and the conclusion was drawn from one dataset. Is there any publicly available droplet-based (10x Genomics) T2D dataset that may demonstrate, and hence validate, this β -cells differentiation block?

Minor:

1. A comprehensive literature review on previously reported AS events identified in T2D was outlined in Paragraph 1-3 of the Introduction. These AS events may serve as positive controls (true AS events) in the analysis. Were these reported AS events detected in the authors' differential splicing analysis?

2. The visual presentation of the PSI values of the AS events were presented in a pseudo-bulk manner, which undermines the advantage of single-cell analysis. For selected genes, the authors demonstrated the gene expression distribution in the form of a violin plot and the individual cells as jittered data points (e.g., Fig3G and I etc.). This was helpful to visualise the gene expression distribution across the single cells. To this end, the PSI profile of AS events originally presented as pseudo-bulk bar plots (e.g., Fig3F and H, Fig 5H and J etc.) may be complemented with violin plots with jittered data points while the pseudo-bulk heatmaps (e.g., Fig 3A, Fig4E etc) may be complemented with heatmap of single cell PSI values.

3. In Fig6C, the authors demonstrated and asserted the decreased in cluster 4 in T2D relative to non-T2D. To formally assess the differences in abundance of all cell clusters between T2D and non-T2D, the authors may use MiloR (<https://github.com/MarioniLab/miloR>; Dann et al., *Nat Biotechnol*, 2021). This software will enable a more formal differential cell abundance testing by returning the test statistics and p-values associated with each cluster.

4. At line 108-110, the authors attributed the poor clustering of Xin dataset relative to Lawlor to lower sequencing depth, and as a consequence, lower number of AS events detected. Another two important factors that may affect the number of AS events detected are read length (150bp, 125/bp etc.) and sequencing mode (single-end, paired-end). It is worth stating the difference/similarity in read length and sequencing mode between these two datasets as potential explanation for lower number of AS events detected in Xin dataset.

5. To ensure the analysis, results, and conclusions of the paper are reproducible, it would be important for the authors to deposit the processed data (R) object and the codes (or R markdown) to reproduce the figures of the manuscript. One such repository that the authors may consider is Zenodo.

Reviewer #2 (Remarks to the Author):

In this study, the authors reanalysed scRNAseq data of islets retrieved from non-diabetic and type 2 diabetic individuals to study the occurrence and contribution of genome-wide alternative splicing to diabetes progression. They have used smart-seq datasets from Lawlor and Xin. There is merit in using smart-seq data for estimating splicing events as it captures full-length mRNA but it is still biased toward 3' end because of oligo dT primers. The authors did not justify why the two datasets were chosen as there are many publicly available healthy and T2D islets datasets. The number of healthy and T2D individuals where the islets were retrieved in the original publications was not indicated. The authors integrated the two datasets (line 98-99) but there was no indication of how this was carried out and how batch effect was corrected, or how normalisation was carried out due to differences in sequencing depth. For this study to be robust, the authors should increase the sample size by integrating more datasets. Importantly, it is not clear which specific splicing profiles or pattern led to beta cell maturation arrest or failure in type 2 diabetes from this study.

Interestingly, SEC13 (an endoplasmic reticulum gene) and C7orf44 gene splicing could identify a subpopulation of beta-cells (fig 2E) but what is not clear is whether this is real or functional, a validation is therefore required.

RNA velocity is used to infer the temporal dynamics or the future states of cells in single cell gene expression data. One of its limitations is that it can produce nonsensical backward velocity flows because it assumes constant transcription rates. Although the authors showed some gene expressions, it is however not clear how they interpreted velocity arrows from 2 to 1 to mean dedifferentiation Fig 7D. How about the arrow from 1 to 10? I think that they have over interpreted RNA velocity. In any case, this also requires orthogonal validation as projections may

be less accurate due to technical noise in single cell data. The concepts of transdifferentiation and dedifferentiation in T2D islets are well documented (PMID: 26713822, PMID: 22980982, PMID: 34068827), I'd have expected this paper to interrogate how the splicing dynamics in beta cells contributed to both scenarios and which pattern is driving them.

Minor:

Line 151-152... the authors indicated that splicing profile define cell type specificity of endocrine cells and referred to fig S4A but what they have shown in their plot (fig S4A) is cell types based on gene expression profile

Reviewer #3 (Remarks to the Author):

In their paper authors use long-read scRNAseq datasets of human islet cells to determine differential exon usage via alternative splicing (AS). The authors suggest non-diabetic islet cells have AS programs that are correlated to cell function and maturity whereas in the case of T2D, this AS profile is lost. They highlight RBPs, hnRNPs, and FXR proteins to deregulate AS in T2D. Although this paper is interesting, its conclusions are highly speculative as the correlation between differential exon usage and the progression of T2D could be explained by aberrant gene expression in T2D or nonsense-mediated RNA decay. Nevertheless, the study is an interesting avenue of T2D biology and warrants study. However, the reviewer has reservations regarding the computational workflow, speculative conclusions, and cell cluster definitions. In its current form, the manuscript would benefit from restructuring/reanalysis in some key areas. Here are some comments that can help strengthen the analysis architecture and thereby improve upon conclusions drawn by the authors.

1. Sharing the vignette of your package and stating "We used the Quantas pipeline" is unacceptable and undermines the process of reproducible computational experiments. As a computational lab I am surprised that the authors would state this without providing a comprehensive analysis outline for this specific paper, this is relevant as the process of moving from an SRA to a Seurat object itself is complex and warrants to the very least a comprehensive github repo. You need to provide a step-by-step and complete coding outline from SRA -> pseudotime trajectory analysis in the form of organized coding scripts, as a github repo maintained by the lead author on this manuscript. It is also good practice (even if the authors don't feel compelled to do so), to save a processed Seurat object, to share with members in the community if requested, particularly since this dataset is not generated by the authors. The coding repo should not just use data in .csv files to reproduce figures but also intrinsically demonstrate the entire analytical strategy utilized especially the differential testing framework.

Not only does this ensure robust reproducibility but is also good for the authors as it allows other inexperienced users to re-run their analysis and reproduce the conclusions drawn in the paper, thereby increasing the possibility of analysis and citation.

2. Point 1 is important because just by looking at the analysis I notice that you have chosen a nominal p-value for statistically significant gene counts, as an example in F5 A-C you state p-value < 0.05 yet in the figure the y-axis is the $-\log_{10}$ of the p-adjusted, which is it? If a coding repository was present for R/python/bash scripts, it would have taken a few seconds to confirm this for the reviewer. All data needs to be FDR corrected using the Benjamini-Hochberg or Benjamini-Yekutieli method(s) and an appropriate padjusted or qvalue needs to be used. Using a nominal pvalue threshold while testing 36601 genes across the hg19/hg38 transcriptome(s) is quite honestly unacceptable. A minimum FDR of $\text{padj} < 0.1$ should be used, while it is more appropriate to use $\text{padj} < 0.05$. For example, in the methods, the authors use an FDR in case of exon quantification yet for genes they use a nominal pvalue. Also interestingly, in the figures the authors state pvalues for exon quantifications such as F3D. The authors need to select one padjusted and use that uniformly across their analysis.

3. The authors state that they identify a median of 4435 and 1528 AS events across the two datasets and analyze them separately. The separate analysis of the two datasets should be a part of the supplemental figures. The authors need to perform an integrated analysis of all cells across the two datasets and establish sets of AS that are unified across multiple cells/datasets/donors (as done in F4, but in place of individual dataset analysis). It is most likely (due to divergent analysis and absent FDR thresholding) that the vast number of AS events are purely dataset specific and therefore it is inappropriate to draw conclusions with regard to T2D progression.

4. Regarding point 3: As a suggestion, the reviewer would benefit from pseudobulking data for each cell type and donor, and then perform DEseq2 perhaps using Limma in case of batch effects (not necessary to use Limma but optional as already DEseq2 can control for batch covariates in the $\sim \text{batch} + \text{condition}$ design formula)

5. F1 should show an integrated UMAP/tSNE across both datasets, along with a UMAP/tSNE for donors/datasets to show optimal cell integration.

6. Contextually subsetting the data is unnecessary (F2), the analysis should be mapped on cells shown in F1A (for integrated, not separate), or if the authors wish to subcluster then this reviewer's question would be how do these clusters map onto endocrine cells when endocrine cells are mapped against other cell types as shown in F1A? The authors need to map their AS profiles onto one unified integrated nonlinear multidimensional plot.

7. Once the authors have achieved point #6, they should then show violin plots for both ND and T2D side by side, the DE analysis for differential AS across cell types should be run on ND only and then used to discern cell type specificity as shown in F53. In any case the profile of a cell type specific AS needs to be split into ND and T2D when shown. The data shown right now in D-F is misleading because the reader can't determine if data shown is ND or T2D.

8. F3B does not show different pathways regulated across cell types, the z-scores are very similar for many gene ontologies. Its fine to show data for similar pathways but also show data for unique pathways perhaps using a dotplot (please review R package clusterprofiler).

9. How do authors consolidate the observation that differential exon usage is not affecting nonsense mediated RNA decay (NMD) F3D? There are only a few genes showing correlation, is NMD defective in T2D? This is probably the most interesting result of the authors, yet they have not elaborated on these gene-AS unions. These genes are probably important for T2D as AS loss in T2D is resulting in NMD or RNA clearance, the authors must try to study these genes closely (but the set needs to be consistent/reproducible across both datasets).

10. 4E Lawlor and Xin is clipped.

11. It is interesting that hnRNPs are downregulated in some datasets but not others, still the reviewer strongly feels that only those results should be discussed which are consistent across datasets. Nevertheless, it is interesting that the splicing machinery in T2D is defective owing to downregulated RNA modulating genes. Perhaps a diagram will help understand the complex interplay of nRNPs, FXRs, FMRs and RNA targets.

12. The authors claim that the depletion of cluster 4 beta cells is possibly what is causing T2D progression and argue that ".....while cluster 4 vanished" line:224-225. This is a misleading claim. F6A-C shows a handful of cells again, restricted to one dataset, and it remains unclear if these cells are truly transcriptionally unique. One must be careful when performing cluster partitioning using k means, as the kmeans can be arbitrarily adjusted and may not reflect biology. This can be evidenced by F6E as there are no statistically different expression levels of any key beta-cell genes, which highlights how clusters 1-4 are most likely a clustering artifact, in fact they are one group of beta cells as evidenced in F6E and F6L conclusively. It is important to demonstrate a DE gene set that differentiates clusters of cells, be as it may, subsets of beta cells. Otherwise claims regarding subsets of beta cells are inappropriate. A useful thought experiment, is that the kmeans can be hypothetically increased till each cell is its own cluster and having its own AS profile, will that be biologically relevant and reproducible across multiple data sets? The authors need to either consolidate beta cells into 1 cluster (similarly for alpha

cells as well), or they need to categorically demonstrate differential activity and gene ontology demarcating cellular function and maturity across subsets of cells. This needs to be performed on this data, not blindly following Dorell's definition of 4 beta cell subtypes.

13. Authors should show the expression of splicing factors for ND and T2D in side-by-side violin plots or dot plots.

14. The authors should generate a heatmap of all RNA species that are affected by FXR1, if FXR1 is truly affecting RNA exon usage, then surely some genes should be exclusively regulated if not, the role of FXR1 being a key AS regulator remains inconclusive.

15. It remains unclear how AS owing to hnRNP dysregulation is leading to beta-alpha cell transition or more so beta cell de-differentiation. Are these hybrid cells (sitting on the boundary of alpha and beta) of high quality similar in median gene expression and mitochondrial content to their alpha and beta counterparts? Such cells can't be detected when staining IF in T2D human tissue, so how do the authors consolidate these discrepancies?

16. In line 299-301 the authors claim their study to be unique in deciphering the role of AS in beta cells and diabetes. This is misleading as numerous studies have demonstrated the role of AS and RBPs in beta cell dysfunction and alternatively the progression of diabetes. (PMID: 31659282, PMID: 36277184, PMID: 34445304, PMID: 33880624, PMID: 36109769, PMID: 32198193). It is appropriate to outline what our colleagues in the field have already established and how this study is unique. The authors should carefully curate pubmed to ensure they highlight their study in view of what is already known.

Point-to-point Response to the Reviewers' Comments

We appreciate the reviewers' time and effort in providing us with comments. These suggestions have largely improved our manuscript. We have substantially revised the manuscript and addressed the reviewers' concerns as below.

Reviewer #1

Single-cell alternative splicing is an underappreciated area of investigation that has the potential to reveal disease and cell type-specific mechanism and biomarkers. The authors characterised the aberrant alternative splicing landscape of endocrine cells derived from type-2 diabetic (T2D) relative to non-T2D controls at single-cell resolution, with particular focus on the insulin-secreting β -cells. I particularly appreciated the demonstration of differentiation block in β -cells revealed through alternative splicing analysis but invisible at the gene expression level. I also appreciated the motif analysis to identify RNA-binding proteins whose motif were enriched among differentially spliced exon, which represents one step further post-differential splicing analysis. Overall, the narrative is concise and logical. While the study lacked experimental or functional validation (such as quantitative polymerase chain reaction, knock-in or knock-out/-down assays) of candidate alternative splicing events, this should not preclude the study from publication because the authors generated probable hypothesis on the disease mechanism and biomarkers from their analysis. However, it is exactly this same point, combined with the fact that majority of conclusions hinged on a single dataset (Lawlor et al.), that the computational analysis and validation should be ensured to be as robust as possible. To this end, I offer the following recommendations. I look forward to the author's response and the imminent publication of this study, provided satisfactory response.

Major:

1. Only skipped-exon (SE) alternative splicing (AS) events were analysed by the authors. Nevertheless, other AS event types have been shown to play important roles in health and disease states, namely mutually exclusive exons (MXE), retained intron (RI), alternative 5' splice site (A5SS), and alternative 3' splice site (A3SS). For example, PAK4 A3SS has been shown to dysregulate insulin and glucagon production (as cited by the

authors in reference number 15). Moreover, two additional AS event types, namely alternative first and last exons (AFE, ALE), have been reviewed to be biologically relevant as well (Alvelos et al., Diabetes Obes Metab, 2018). In the interest of a more comprehensive characterisation of the AS landscape in T2D and also in identifying additional and novel biomarkers, the authors should include all seven AS event types in the analysis.

Re: Thank you for the comments. We identified and investigated all seven AS event types (SE, MXE, RI, A5SS, A3SS, AFE, and ALE) generated by MARVEL software and Quantas (**Revised Fig. S17-S18**). Both methods consistently identified SE as the predominant alternative splicing event, followed by MXE and A3SS as the second- and third-most prevalent types according to Quantas, and A3SS and AFE according to MARVEL (**Revised Fig. S17A-B**).

To figure out additional biomarkers, we performed clustering based on these AS types (SE, MXE, A3SS, AFE), but we observed that MXE, A3SS, AFE failed to clearly define the major endocrine cell types (**Revised Fig. S17C-H**). We further included SE with the other six AS event types for clustering, but they did not significantly improve the clustering resolution compared to using SE only (**Revised Fig. S17I-J**), due to the lower AS event numbers of the other six AS event types. In addition, SE contributes to the most influential biomarkers in β -cells when identifying cell-specific AS events, including seven AS types (**Revised Fig. S18**). This limitation has also been presented in the published report which concentrated on cassette exon (SE) only in the single-cell splicing analysis ¹. It might remain a challenge to quantify adequate events for these six AS types for single-cell analysis. Therefore, we also focused on the cassette exons (SE) representing the primary type of AS events. We added these results and discussion in the revised version (**Line 373 on Page 11**).

2. Quantas pipeline was used to detect, quantify, and perform differential splicing analysis. Quantas was first developed in 2013 originally for AS analysis of bulk RNAseq. Since then, several specialised single-cell AS analysis softwares were developed, benchmarked, and peer-reviewed. These published single-cell AS analysis softwares demonstrated their ability to comprehensively characterise the single-cell AS landscape

and to address challenges specific to AS analysis in single cells. One such published software is MARVEL (<https://github.com/wenweixiong/MARVEL>; Wen et al., Nucleic Acid Res, 2023). It would be of particular importance to orthogonally validate the differentially spliced AS events detected by Quantas by identify AS events that are detected by both approaches and missed AS events uniquely detected by either approach.

Re: As suggested, we compared the AS events detected by both MARVEL and Quantas. First, both methods consistently identified SE as the predominant alternative splicing event, followed by MXE and A3SS as the second- and third-most prevalent types according to Quantas, and A3SS and AFE according to MARVEL (**Revised Fig. S17A-B**). Then we performed the clustering based on SE detected by MARVEL and found it showed a similar clustering pattern compared to the result by Quantas, which enabled us to define α -cells, β -cells, and their subsets, but not other pancreatic cells (**Revised Fig. 1C-D and S17C-D**). We did observe that Quantas identified considerably more MXE, A3SS and RI, but none of ALE and AFE, compared to MARVEL (**Revised Fig. S17A-B**), and Quantas identified considerably more AS markers of beta-cells compared to MARVEL (**Revised Fig. S18**), indicating the difference in identifying the other six AS types and for differential analysis.

However, we found two methods that can reveal the consistent differential tendency of some SE splicing events which we are concerned about in our study (**Revised Fig. 3F, 3H, 4F, and 4G**).

3. Given the lack of experimental validation of candidate splicing AS events, it would be of particular importance to perform in silico visual-based validation of these AS events. The authors performed in silico validation in a pseudo-bulk manner, but this undermines the advantage of single-cell analysis (Fig4F and G). One such software that enables single-cell in silico validation of AS events is VALERIE (<https://github.com/wenweixiong/VALERIE>; Wen et al., PLoS Comput Biol, 2020). in silico validation will lend confidence for selection of true positive AS events for the postulation of disease mechanism and downstream experimental validation.

Re: Thanks for the suggestion. We validated our analysis using VALERIE and added these results in the revised version (**Revised Fig. 3F, 3H, 4F, and 4G**).

4. The β -cells differentiation block demonstrated by the authors were particularly intriguing and indicated a mechanism for T2D with therapeutic implications. The RNA velocity analysis was convincing, but cell numbers were nevertheless small, and the conclusion was drawn from one dataset. Is there any publicly available droplet-based (10x Genomics) T2D dataset that may demonstrate, and hence validate, this β -cells differentiation block?

Re: To validate the observation of T2D β -cells differentiation block, we incorporated three independent public datasets (the Lawlor dataset, the Xin dataset, and the Fang dataset), and applied three independent trajectory methods (CytoTrace, mature scores, Monocle 2). We observed the mature β -cells in T2D were greatly decreased in all three datasets (**Revised Fig. 6I-J, 7E and S14**), robustly indicating T2D β -cells differentiation arrest.

Minor:

5. A comprehensive literature review on previously reported AS events identified in T2D was outlined in Paragraph 1-3 of the Introduction. These AS events may serve as positive controls (true AS events) in the analysis. Were these reported AS events detected in the authors' differential splicing analysis?

Re: We did detect significantly different splicing in VEGF between T2D and ND but did not detect GCK and PAX exon junctions. This may be attributed to the limitations in sequencing depth and inadequate exon junctions.

6. The visual presentation of the PSI values of the AS events were presented in a pseudo-bulk manner, which undermines the advantage of single-cell analysis. For selected genes, the authors demonstrated the gene expression distribution in the form of a violin plot and the individual cells as jittered data points (e.g., Fig3G and I etc.). This was helpful to visualise the gene expression distribution across the single cells. To this end, the PSI profile of AS events originally presented as pseudo-bulk bar plots (e.g., Fig3F and H, Fig 5H and J etc.) may be complemented with violin plots with jittered data points while the pseudo-bulk heatmaps (e.g., Fig 3A, Fig4E etc) may be complemented with heatmap of single cell PSI values.

Re: Thanks for the suggestion. We visualized these gene-splicing levels with single-cell heatmaps and presented mean PSI across the genomic coordinate using VALERIE (**Revised Fig. 3F, 3H, 4F, and 4G**).

7. In Fig6C, the authors demonstrated and asserted the decreased in cluster 4 in T2D relative to non-T2D. To formally assess the differences in abundance of all cell clusters between T2D and non-T2D, the authors may use MiloR (<https://github.com/MarioniLab/miloR>; Dann et al., Nat Biotechnol, 2021). This software will enable a more formal differential cell abundance testing by returning the test statistics and p-values associated with each cluster.

Re: We have replaced it with the output of MiloR (**Revised Fig. 6C**).

8. At line 108-110, the authors attributed the poor clustering of Xin dataset relative to Lawlor to lower sequencing depth, and as a consequence, lower number of AS events detected. Another two important factors that may affect the number of AS events detected are read length (150bp, 125/bp etc.) and sequencing mode (single-end, paired-end). It is worth stating the difference/similarity in read length and sequencing mode between these two datasets as potential explanation for lower number of AS events detected in Xin dataset.

Re: We discussed it in the revision (**Line 112 on Page 4 and Line 401 on Page 12**).

9. To ensure the analysis, results, and conclusions of the paper are reproducible, it would be important for the authors to deposit the processed data (R) object and the codes (or R markdown) to reproduce the figures of the manuscript. One such repository that the authors may consider is Zenodo.

Re: We have uploaded our codes and the processed data to GitHub (https://github.com/Helen-weshy1/AS_project) and declared them in Methods.

Reviewer #2

1. In this study, the authors reanalysed scRNAseq data of islets retrieved from non-diabetic and type 2 diabetic individuals to study the occurrence and contribution of genome-wide alternative splicing to diabetes progression. They have used smart-seq

datasets from Lawlor and Xin. There is merit in using smart-seq data for estimating splicing events as it captures full-length mRNA but it is still biased toward 3' end because of oligo dT primers.

Re: Thanks for the comments. We clarified the limitations of smart-based sequencing in Discussion of the revised version (**Line 379 on Page 11**).

2. The authors did not justify why the two datasets were chosen as there are many publicly available healthy and T2D islets datasets. The number of healthy and T2D individuals where the islets were retrieved in the original publications was not indicated.

Re: We chose the Lawlor and Xin data for our analysis in the manuscript after evaluating seven independent scRNA-seq datasets of human islets, of which five are full-length RNA-seq (**Revised Table S1**). Given that single-cell AS events are much higher sparse matrix than the gene expression matrix, we prioritized the higher depth of the datasets, which largely impacts the clustering (**Line 112 on Page 4 and Line 397 on Page 12**), with full-length RNA-seq, which allowed us to acquire a satisfactory number of quantifiable exon junctions to characterize cell populations at the single-cell level. Cell numbers also should be considered to ensure the identification of cell subsets. The Lawlor data as the first option showed high depth and satisfactory cell numbers with full-length RNA-seq, and the second was the Xin data. We clarified this in the revised version (**Line 81 on Page 3**).

We summarized these evaluated datasets including cells they retrieved in their publications, sequence depth, read length, and sequence modes as well. (**Revised Table S1**).

3. The authors integrated the two datasets (line 98-99) but there was no indication of how this was carried out and how batch effect was corrected, or how normalization was carried out due to differences in sequencing depth. For this study to be robust, the authors should increase the sample size by integrating more datasets.

Re: Thanks for pointing it out. We incorporated diverse independent human datasets but did not directly combine these data because there were no appropriate normalization methods and batch effect correction methods for AS event profiles or Psi matrix. So, as

the strategy described in the publication ¹, we analyzed these datasets separately and then integrated their results and conclusions.

To derive robust findings in our study, we presented the observation of commonality to generate conclusions from the analysis of multiple independent datasets by employing a variety of methods. First, we utilized MARVEL to perform clustering which exhibited a similar pattern to using the Quantas pipeline (**Revised Fig. 1C-D and S17C-D**), validating that cell types can be defined by AS events with different algorithms. Furthermore, we confirmed the observation that AS regulation in endocrine cells is an orthogonal layer of gene expression in two smart-seq datasets (the Lawlor dataset, the Xin dataset) (**Revised Fig. 2D-E, 3C-I and S4**), which was also observed in neuron development and hematopoiesis ^{1,2}. Third, we further utilized VALERIE to present and evaluate AS differential analysis (**Revised Fig. 3F, 3H, 4F, and 4G**), revealing consistent robustness with Quantas method. Fourth, by incorporating three independent public datasets (the Lawlor dataset, the Xin dataset, and the Fang dataset), and applying three independent trajectory methods (CytoTRACE, maturity scores and monocle 2), we orthogonally validated β -cell maturation arrest in T2D in all the datasets with all the methods (**Revised Fig. 6I-J, 7E and S14**). Fifth, we presented AS profiles associated with the regulations in T2D β -cell based on the common dynamic splicing profiles of two smart-seq datasets, but not the dataset-specific profiles (**Revised Fig. 4**). In addition, we validated the β -cell dedifferentiation and transdifferentiation results by integrating RNA velocity flows and monocle 2 trajectory analysis (**Revised Fig. 7D-E**). Therefore, all these incorporations of independent datasets and methods revealed consistently robust conclusions of this study (**Line 360 on Page 11**).

4. Importantly, it is not clear which specific splicing profiles or pattern led to beta cell maturation arrest or failure in type 2 diabetes from this study.

Re: Thanks for the comments. By comparing healthy and T2D β -cells, as well as the mature and immature β -cell subpopulations, we identified “RNA splicing”, “autophagy”, “cell cycle”, “nucleotide triphosphate metabolic process” and “mitochondrial inner membrane” gene aberrant splicing may lead to β -cell maturation arrest (**Revised Fig 4A-**

D, 6L, and S15-S16). We summarized the results in the revised version (**Line 258 on Page 8 and Line 347 on Page 10**).

5. Interestingly, SEC13 (an endoplasmic reticulum gene) and C7orf44 gene splicing could identify a subpopulation of beta-cells (fig 2E) but what is not clear is whether this is real or functional, a validation is therefore required.

Re: Thanks for this important suggestion. Exploring the functionality of these markers would add an intriguing dimension to the analysis, addressing the question of whether they actively contribute to β -cell functionality or serve merely as identifiers for specific subsets. The role of SEC13 in regulating endoplasmic reticulum (ER) membrane formation and vesicle bud formation ³ may suggest its potential involvement in β -cell maturation and insulin secretion. C7orf44, identified as a pseudogene, lacks comprehensive investigation. Functional validation would be one of the important future directions in our research.

6. RNA velocity is used to infer the temporal dynamics or the future states of cells in single cell gene expression data. One of its limitations is that it can produce nonsensical backward velocity flows because it assumes constant transcription rates. Although the authors showed some gene expressions, it is however not clear how they interpreted velocity arrows from 2 to 1 to mean dedifferentiation Fig 7D. How about the arrow from 1 to 10? I think that they have over interpreted RNA velocity. In any case, this also requires orthogonal validation as projections may be less accurate due to technical noise in single cell data.

Re: Thank you for this important comment. We employed Monocle 2 to carefully validate the velocity flows. Monocle 2 reconstructed trajectories from healthy and T2D beta-cells with three and two branches, respectively (**Revised Fig. 7E**). Cluster 1 was concentrated at the top of the tree, and partial cells of cluster 2 and 3 were distributed over the remainder of the tree in healthy cells (**Revised Fig. 7E**), indicating the differentiation from cluster 1 to cluster 2, whereas more cells of cluster 2 were concentrated at the top and less at the bottom compared to cells of cluster 1 in T2D (**Revised Fig. 7E**), indicating the dedifferentiation from cluster 2 to cluster 1, which was supported by velocity analysis. For the velocity arrow from cluster 1 to cluster 10, the marker gene expression could not well

explain this trajectory, meanwhile cluster 10 mixed multiple endocrine cell types (**Revised Fig. 2C**). Therefore, we agree that the velocity flow from 1 to 10 might be nonsensical and we have toned down the results and descriptions in the revised version.

7. The concepts of transdifferentiation and dedifferentiation in T2D islets are well documented (PMID: 26713822, PMID: 22980982, PMID: 34068827), I'd have expected this paper to interrogate how the splicing dynamics in beta cells contributed to both scenarios and which pattern is driving them.

Re: Thanks for the suggestion. We investigated the splicing profiles of the β -cell subsets that were involved in transdifferentiation (cluster 3) and dedifferentiation (cluster 2) in T2D (**Revised Fig. S16**), combining the differential analysis of ND and T2D β -cells (**Revised Fig. 4**). In summary, we found altered splicing diversity, energy status and cell cycle at AS regulation layers are associated with regulation of β -cell transdifferentiation and dedifferentiation (**Revised Fig. 4 and S16**). We added the analysis and descriptions in the revision (**Line 347 on Page 10**).

Minor:

8. Line 151-152... the authors indicated that splicing profile define cell type specificity of endocrine cells and referred to fig S4A but what they have shown in their plot (fig S4A) is cell types based on gene expression profile.

Re: We have fixed it.

Reviewer #3

In their paper authors use long-read scRNAseq datasets of human islet cells to determine differential exon usage via alternative splicing (AS). The authors suggest non-diabetic islet cells have AS programs that are correlated to cell function and maturity whereas in the case of T2D, this AS profile is lost. They highlight RBPs, hnRNPs, and FXR proteins to deregulate AS in T2D. Although this paper is interesting, its conclusions are highly speculative as the correlation between differential exon usage and the progression of T2D could be explained by aberrant gene expression in T2D or nonsense-mediated RNA decay. Nevertheless, the study is an interesting avenue of T2D biology and warrants

study. However, the reviewer has reservations regarding the computational workflow, speculative conclusions, and cell cluster definitions. In its current form, the manuscript would benefit from restructuring/reanalysis in some key areas. Here are some comments that can help strengthen the analysis architecture and thereby improve upon conclusions drawn by the authors.

1. Sharing the vignette of your package and stating “We used the Quantas pipeline” is unacceptable and undermines the process of reproducible computational experiments. As a computational lab I am surprised that the authors would state this without providing a comprehensive analysis outline for this specific paper, this is relevant as the process of moving from an SRA to a Seurat object itself is complex and warrants to the very least a comprehensive github repo. You need to provide a step-by-step and complete coding outline from SRA -> pseudotime trajectory analysis in the form of organized coding scripts, as a github repo maintained by the lead author on this manuscript. It is also good practice (even if the authors don't feel compelled to do so), to save a processed Seurat object, to share with members in the community if requested, particularly since this dataset is not generated by the authors. The coding repo should not just use data in .csv files to reproduce figures but also intrinsically demonstrate the entire analytical strategy utilized especially the differential testing framework. Not only does this ensure robust reproducibility but is also good for the authors as it allows other inexperienced users to re-run their analysis and reproduce the conclusions drawn in the paper, thereby increasing the possibility of analysis and citation.

Re: Thank you for the comment. We have uploaded our step-by-step codes and necessary processed data to GitHub (https://github.com/Helen-weshy1/AS_project) and declared the code availability in the revised manuscript.

2. Point 1 is important because just by looking at the analysis I notice that you have chosen a nominal p-value for statistically significant gene counts, as an example in FS5 A-C you state p-value <0.05 yet in the figure the y-axis is the -log₁₀ of the p-adjusted, which is it? If a coding repository was present for R/python/bash scripts, it would have taken a few seconds to confirm this for the reviewer. All data needs to be FDR corrected using the Benjamini-Hochberg or Benjamini-Yekutieli method(s) and an appropriate

padjusted or qvalue needs to be used. Using a nominal pvalue threshold while testing 36601 genes across the hg19/hg38 transcriptome(s) is quite honestly unacceptable. A minimum FDR of $\text{padj} < 0.1$ should be used, while it is more appropriate to use $\text{padj} < 0.05$. For example, in the methods, the authors use an FDR in case of exon quantification yet for genes they use a nominal pvalue. Also interestingly, in the figures the authors state pvalues for exon quantifications such as F3D. The authors need to select one padjusted and use that uniformly across their analysis.

Re: Thank you for pointing it out. The p -values in our manuscript are all adjusted by FDR, not nominal p -values. We clarified it in the revised manuscript.

3. The authors state that they identify a median of 4435 and 1528 AS events across the two datasets and analyze them separately. The separate analysis of the two datasets should be a part of the supplemental figures. The authors need to perform an integrated analysis of all cells across the two datasets and establish sets of AS that are unified across multiple cells/datasets/donors (as done in F4, but in place of individual dataset analysis). It is most likely (due to divergent analysis and absent FDR thresholding) that the vast number of AS events are purely dataset specific and therefore it is inappropriate to draw conclusions with regard to T2D progression.

Re: Thanks for the comment. We incorporated diverse independent human datasets but did not directly combine these data because there were no appropriate normalization methods and batch effect correction methods for AS event profiles or Psi matrix. So, as the strategy described in the publication ¹, we analyzed these datasets separately and then integrated their results and conclusions.

It has been well described that independent human pancreatic single-cell data from diabetic individuals showed great discrepancy because of the complex etiology of disease ⁴. However, they are still available to reveal common regulations during diabetes. Importantly, we discovered and presented the observations of commonality which have been validated in each dataset with multiple methods, to generate robust and consistent conclusions in these data, although the large number of dataset-specific observations can be found.

We summarized the integrated analysis and conclusions of commonality from multiple datasets in the reply to Q#3 of Reviewer #2.

4. Regarding point 3: As a suggestion, the reviewer would benefit from pseudobulking data for each cell type and donor, and then perform DESeq2 perhaps using Limma in case of batch effects (not necessary to use Limma but optional as already DESeq2 can control for batch covariates in the \sim batch + condition design formula).

Re: Thank you for the suggestion. We found DESeq2 generated a similar conclusion compared to the differential gene expression results by Quantas. That's very helpful suggestions.

5. F1 should show an integrated UMAP/tSNE across both datasets, along with a UMAP/tSNE for donors/datasets to show optimal cell integration.

Re: As suggested, we integrated both datasets with UMAP as below (**Response Fig. 1A**), and annotated cell types by hormone expression (**Response Fig. 1B**). The integrated data clustered the subpopulations identical to the separated data.

Response Fig. 1. UMAP plots of integrated the Lawlor data and the Xin data

6. Contextually subsetting the data is unnecessary (F2), the analysis should be mapped on cells shown in F1A (for integrated, not separate), or if the authors wish to subcluster then this reviewer's question would be how do these clusters map onto endocrine cells when endocrine cells are mapped against other cell types as shown in F1A? The authors need to map their AS profiles onto one unified integrated nonlinear multidimensional plot.

Re: As suggested, we performed AS clustering without subsetting (**Revised Figure 1C-D**). We tried to map AS profiles onto the integrated gene expression UMAP and tSNE plots (**Response Fig. 2**). However, the AS clusters in the gene expression plot could not be clearly defined (**Response Fig. 2**). So, we adopted the strategy to map gene expression profiles on AS clustering plot as described in the publication ¹ (**Revised Fig. 1C-D**). In addition, we incorporated diverse independent human datasets but did not directly combine these data because there were no appropriate normalization methods and batch effect correction methods for AS event profiles or Psi matrix. So, as the strategy described in the publication ¹, we analyzed these datasets separately and then integrated their results and conclusions.

Response Fig. 2. AS profiles mapping onto the integrated gene expression UMAP (A) and tSNE (B) plots

7. Once the authors have achieved point #6, they should then show violin plots for both ND and T2D side by side, the DE analysis for differential AS across cell types should be run on ND only and then used to discern cell type specificity as shown in FS3. In any case the profile of a cell type specific AS needs to be split into ND and T2D when shown. The

data shown right now in D-F is misleading because the reader can't determine if data shown is ND or T2D.

Re: As suggested, before that, we run on ND only and then T2D in beta-cells, and we found ND and T2D β -cells shared only around 15% cell-type specific AS events by separating analysis (**Revised Fig. S6**), indicating the great difference of ND and T2D beta-cell AS profiles. Therefore, we further performed the AS differential analysis to figure out AS differential analysis between ND and T2D cells (**Revised Fig. 4**).

We clarified that Fig. 2D-F showed merged ND and T2D data to show the AS markers for each subset.

8. F3B does not show different pathways regulated across cell types, the z-scores are very similar for many gene ontologies. Its fine to show data for similar pathways but also show data for unique pathways perhaps using a dotplot (please review R package clusterprofiler).

Re: We replaced it with the top presentative gene ontologies of each cell type instead of numerous gene ontologies, which showed distinguished z-scores (**Revised Fig. 3B**). We appreciated your suggestion to use dot plots in several figures to present our data (**Revised Fig. 6E, S10 and S16**).

9. How do authors consolidate the observation that differential exon usage is not affecting nonsense mediated RNA decay (NMD) F3D? There are only a few genes showing correlation, is NMD defective in T2D? This is probably the most interesting result of the authors, yet they have not elaborated on these gene-AS unions. These genes are probably important for T2D as AS loss in T2D is resulting in NMD or RNA clearance, the authors must try to study these genes closely (but the set needs to be consistent/reproducible across both datasets).

Re: Thanks for the suggestion. The activation of the NMD pathway might be associated with proinflammatory levels in diabetes ⁵. We investigated the NMD gene expression in both the Lawlor and the Xin datasets. We did not find significant differential expression between T2D and ND beta-cells (**Response Fig. 3**). More experimental evidence remains to be needed to facilitate the understanding of NMD and T2D.

Response Fig. 3. NMD gene expression in ND and T2D β -cells

10. 4E Lawlor and Xin is clipped.

Re: We have fixed it.

11. It is interesting that hnRNPs are downregulated in some datasets but not others, still the reviewer strongly feels that only those results should be discussed which are consistent across datasets. Nevertheless, it is interesting that the splicing machinery in T2D is defective owing to downregulated RNA modulating genes. Perhaps a diagram will help understand the complex interplay of nRNPs, FXRs, FMRs and RNA targets.

Re: We added more description (**Line 335 on Page 10**) and generated a diagram here to illustrate how downregulated RBPs regulate target gene splicing as below (**Response Fig. 4**).

Response Fig. 4. A diagram of interplay of RBPs and their targets

12. The authors claim that the depletion of cluster 4 beta cells is possibly what is causing T2D progression and argue that “.....while cluster 4 vanished” line:224-225. This is a misleading claim. F6A-C shows a handful of cells again, restricted to one dataset, and it remains unclear if these cells are truly transcriptionally unique. One must be careful when performing cluster partitioning using k means, as the kmeans can be arbitrarily adjusted and may not reflect biology. This can be evidenced by F6E as there are no statistically different expression levels of any key beta-cell genes, which highlights how clusters 1-4 are most likely a clustering artifact, in fact they are one group of beta cells as evidenced in F6E and F6L conclusively. It is important to demonstrate a DE gene set that differentiates clusters of cells, be as it may, subsets of beta cells. Otherwise claims regarding subsets of beta cells are inappropriate. A useful thought experiment, is that the kmeans can be hypothetically increased till each cell is its own cluster and having its own AS profile, will that be biologically relevant and reproducible across multiple data sets? The authors need to either consolidate beta cells into 1 cluster (similarly for alpha cells as well), or they need to categorically demonstrate differential activity and gene ontology demarcating cellular function and maturity across subsets of cells. This needs to be performed on this data, not blindly following Dorell’s definition of 4 beta cell subtypes.

Re: Thanks for the comments. We used the SNN algorithm for clustering, but not k-means. Furthermore, we defined clusters 1 to 4 based on exon included levels (Psi matrix) but

not based on gene expression, and we found hundreds of included/skipped splicing markers for each cluster (**Revised Fig S11**). However, we still found a bunch of gene expression markers for each cluster which can also clearly separate clusters 1 to 4 (**Revised Fig. 6D**), although the number of gene expression markers are less than AS markers (**Revised Fig S11 and S12**). Moreover, clusters 1 to 4 can be defined based on beta-cell mature gene expression and CytoTRACE, exhibiting higher mature gene expression levels in cluster 4 than in other clusters (**Revised Fig 6G-H**). Additionally, based on gene expression, Monocle 2 also well defined clusters 1 to 4, which distributed along the developmental trajectory in different stages (**Revised Fig 6I-J**). These results demonstrate that clusters 1 to 4 are not transcriptionally unique. These subsets of β -cells can be separated not only at splicing levels, but also at gene expression levels.

13. Authors should show the expression of splicing factors for ND and T2D in side-by-side violin plots or dot plots.

Re: We fixed it.

14. The authors should generate a heatmap of all RNA species that are affected by FXR1, if FXR1 is truly affecting RNA exon usage, then surely some genes should be exclusively regulated if not, the role of FXR1 being a key AS regulator remains inconclusive.

Re: As suggested, we generated heatmaps of all potential targeted exons that contained FXR1 splice site (A[CT]GAC[AG]) that might be affected by FXR1 (**Response Fig. 5**). We did not observe that FXR1 expression significantly correlated with these exons included, but we still cannot exclude the AS regulation of FXR1 in β -cells since we cannot simply speculate all authentic targets of FXR1. The predicted splice site could be the potentially required for the target gene splicing, which does not mean any RNA with this motif is sufficient to be spliced, because the sequence contexts of splice sites impose idiosyncratic constraints on the recognition of alternative splice sites by the core splicing machinery, and the splicing process is a cooperativity between plenty of trans- and cis-factors (Fu, Nat Rev Genet 2015). A reliable strategy would be utilizing CLIP-seq to evaluate the FXR1 targets and using genetic models to validate its functions in β -cells, which would be one of the important future directions in our research. Thus, in our

manuscript, we stated FXR1 is associated with exon splicing and β -cell maturation which may play a potential regulator of β -cell function.

Response Fig. 5. Heatmap showing all potential targeted exons that contained FXR1 splice site (A[CT]GAC[AG]), which ranked by their correlation with FXR1 expression

15. It remains unclear how AS owing to hnRNP dysregulation is leading to beta-alpha cell transition or more so beta cell de-differentiation. Are these hybrid cells (sitting on the boundary of alpha and beta) of high quality similar in median gene expression and mitochondrial content to their alpha and beta counterparts? Such cells can't be detected when staining IF in T2D human tissue, so how do the authors consolidate these discrepancies?

Re: We did not directly remove these dual hormonal cells in the dataset because of previous reports in human pancreatic tissue⁶⁻⁹. We noted that Spijker et al., 2015 reported an increase of dual hormone (INS/GCG) cells in human T2D whereas the study by van Gurp et al., 2022 showed a comparable frequency^{6,7}. Although the investigation of these cells in human pancreatic islets is controversial in diabetes, we carefully performed quality control of *INS/GCG* dual hormonal cells with other cell types including α -cells and β -cells counterparts (**Revised Fig. S2B-D**). We observed that these dual hormonal cells displayed comparable quality with other cell types, which is why we opted not to exclude them from the datasets. Nevertheless, we did not perform deep analysis of these cells due to their very limited cell mass and complicated roles in pancreatic tissue.

16. In line 299-301 the authors claim their study to be unique in deciphering the role of AS in beta cells and diabetes. This is misleading as numerous studies have demonstrated the role of AS and RBPs in beta cell dysfunction and alternatively the progression of diabetes. (PMID: 31659282, PMID: 36277184, PMID: 34445304, PMID: 33880624, PMID: 36109769, PMID: 32198193). It is appropriate to outline what our colleagues in the field have already established and how this study is unique. The authors should carefully curate pubmed to ensure they highlight their study in view of what is already known.

Re: We have cited these publications and rewritten the description and in our revised manuscript.

Reference

- 1 Feng, H. *et al.* Complexity and graded regulation of neuronal cell-type-specific alternative splicing revealed by single-cell RNA sequencing. *Proc Natl Acad Sci U S A* **118**, doi:10.1073/pnas.2013056118 (2021).
- 2 Wang, F. *et al.* Single-cell architecture and functional requirement of alternative splicing during hematopoietic stem cell formation. *Sci Adv* **8**, eabg5369, doi:10.1126/sciadv.abg5369 (2022).
- 3 Budnik, A. & Stephens, D. J. ER exit sites - Localization and control of COPII vesicle formation. *Febs Lett* **583**, 3796-3803, doi:10.1016/j.febslet.2009.10.038 (2009).
- 4 Wang, Y. J. & Kaestner, K. H. Single-Cell RNA-Seq of the Pancreatic Islets--a Promise Not yet Fulfilled? *Cell Metab* **29**, 539-544, doi:10.1016/j.cmet.2018.11.016 (2019).
- 5 Ghiasi, S. M. & Rutter, G. A. Consequences for Pancreatic beta-Cell Identity and Function of Unregulated Transcript Processing. *Front Endocrinol (Lausanne)* **12**, 625235, doi:10.3389/fendo.2021.625235 (2021).
- 6 Spijker, H. S. *et al.* Loss of β -Cell Identity Occurs in Type 2 Diabetes and Is Associated With Islet Amyloid Deposits. *Diabetes* **64**, 2928-2938, doi:10.2337/db14-1752 (2015).
- 7 van Gurp, L. *et al.* Generation of human islet cell type-specific identity genesets. *Nat Commun* **13**, 2020, doi:10.1038/s41467-022-29588-8 (2022).
- 8 Enge, M. *et al.* Single-Cell Analysis of Human Pancreas Reveals Transcriptional Signatures of Aging and Somatic Mutation Patterns. *Cell* **171**, 321-330 e314, doi:10.1016/j.cell.2017.09.004 (2017).
- 9 Blodgett, D. M. *et al.* Novel Observations From Next-Generation RNA Sequencing of Highly Purified Human Adult and Fetal Islet Cell Subsets. *Diabetes* **64**, 3172-3181, doi:10.2337/db15-0039 (2015).

Reviewers' comments:

Reviewer #1 (Remarks to the Author):

The authors have addressed my comments satisfactorily. I have no further comments. Well done.

Reviewer #2 (Remarks to the Author):

The authors have sufficiently addressed my concerns. The paper is in a well improved state

Reviewer #3 (Remarks to the Author):

The authors have made efforts to address the comments provided by the reviewers. However, some outstanding concerns that can not be ignored remain.

1. While individual R scripts are annotated, The Github repo description is missing altogether. The repo lacks a basic description or README.md file which ties all of the individual scripts together. We should not be having this back and forth, the authors need to establish a well-curated repo with expansive descriptions, or at the very least a well-curated README.md (introduction to the data) file explaining what these analyses and subfiles are. Do the authors feel that navigating the existing repo is intuitive? I hope we will not have to discuss GitHub repos again. Github repos are like computational methods sections, we need to do better at curating them appropriately to allow for robust reproducibility.

2. In the response to reviewers, the authors state: "As suggested, we generated heatmaps of all potential targeted exons that contained FXR1 splice site (A[CT]GAC[AG]) that might be affected by FXR1 (Response Fig. 5). We did not observe that FXR1 expression significantly correlated with these exons included, but we still cannot exclude the AS regulation of FXR1 in beta-cells since we cannot simply speculate all authentic targets of FXR1." If this is not an authentic target then why study it at all? The FXR1 motif is well-defined owing to studying its implications in cancer and thus defined splice FXR1 sites between exons exist, thus if genes containing these splice sites are not affected then the role of FXR1 as a splicing regulator is speculative. The reasoning provided by the authors is also speculative. If we are to speculate then FXR1 works on modulating translation via FMRP/CYFIP1/eIF4E So what is ground truth? Thus lines (355-358) are speculative and inappropriate. The authors should make a list of exons (at least 1 per gene, if multiple motifs exist/gene) affected by FXR1, plot a heatmap (for exons showing significant changes say by 1.5x) alongside a pseudobulk gene expression heatmap of the corresponding gene across ND and T2D donors, and include it in the paper. This will result in unbiased data reporting. Also, at the very least the authors need to make sure the discussion reflects these speculations and deficiencies.

3. Include the NMD gene expression as a heatmap, and add it as a figure in the supplemental. Perhaps run a cell-by-cell statistical analysis and add asterisks where significant genes exist.

Point-to-point Response to the Reviewers' Comments

We appreciate the reviewers' time and effort in providing us with comments. These suggestions have largely improved our manuscript. We have substantially revised the manuscript and addressed the reviewers' concerns as below.

Reviewer #3

1. While individual R scripts are annotated, The Github repo description is missing altogether. The repo lacks a basic description or README.md file which ties all of the individual scripts together. We should not be having this back and forth, the authors need to establish a well-curated repo with expansive descriptions, or at the very least a well-curated README.md (introduction to the data) file explaining what these analyses and subfiles are. Do the authors feel that navigating the existing repo is intuitive? I hope we will not have to discuss GitHub repos again. Github repos are like computational methods sections, we need to do better at curating them appropriately to allow for robust reproducibility.

Re: Thank the reviewer for the comments and suggestions, which is highly appreciated. We have uploaded the *README.md* file accordingly to GitHub (https://github.com/Helenweshy1/AS_project).

2. In the response to reviewers, the authors state: "As suggested, we generated heatmaps of all potential targeted exons that contained FXR1 splice site (A[CT]GAC[AG]) that might be affected by FXR1 (Response Fig. 5). We did not observe that FXR1 expression significantly correlated with these exons included, but we still cannot exclude the AS regulation of FXR1 in beta-cells since we cannot simply speculate all authentic targets of FXR1." If this is not an authentic target then why study it at all? The FXR1 motif is well-defined owing to studying its implications in cancer and thus defined splice FXR1 sites between exons exist, thus if genes containing these splice sites are not affected then the role of FXR1 as a splicing regulator is speculative. The reasoning provided by the authors is also speculative. If we are to speculate then FXR1 works on modulating translation via FMRP/CYFIP1/eIF4E So what is ground truth? Thus lines (355-358) are

speculative and inappropriate. The authors should make a list of exons (at least 1 per gene, if multiple motifs exist/gene) affected by FXR1, plot a heatmap (for exons showing significant changes say by 1.5x) alongside a pseudobulk gene expression heatmap of the corresponding gene across ND and T2D donors, and include it in the paper. This will result in unbiased data reporting. Also, at the very least the authors need to make sure the discussion reflects these speculations and deficiencies.

Re: Thank the reviewer for the comments and suggestions. We have identified potentially targeted exons of FXR1 and analyzed the correlation of these exon inclusion levels and FXR1 mRNA expression levels across β -cells in the revised manuscript (**Revised Fig. 6O** and **Fig. S16**). The description was added in Line 264: “We identified 79 exons or their flanking sequence from 70 genes containing potential FXR1 binding motifs across β -cells, with 29 of them described previously (George, J. et al., *Cell Rep* 37, 109934,2021). Gene expression levels of FXR1 correlated with inclusion levels of these exons, suggesting that higher FXR1 expression levels are linked to higher exon inclusion levels in mature β -cells, whereas lower FXR1 expression levels with lower exon inclusion levels in immature cells (*Cell Rep* 37, 109934,2021). Interestingly, β -cell subsets exhibited cluster-specific targeted exon inclusion, indicating the diverse targets of FXR1 at different stages of β -cell maturation (**Fig. 6O** and **Fig. S16A**).”

3. Include the NMD gene expression as a heatmap, and add it as a figure in the supplemental. Perhaps run a cell-by-cell statistical analysis and add asterisks where significant genes exist.

Re: Thank the reviewer for the suggestions. We have added a heatmap and a violin plot describing NMD gene expression levels in the revised manuscript (**Revised Fig. S18**), which is described in the *DISCUSSION* section (**Line 367**). None of these genes showed significant different expression levels between ND and T2D β -cells, as we described in the text.

REVIEWERS' COMMENTS:

Reviewer #3 (Remarks to the Author):

The authors have sufficiently addressed my concerns. Thankfully the Github repo is now acceptable.
Thank you.